# ELIMINATING AGENTIC WORKFLOW FOR INTRODUCTION GENERATION WITH PARAMETRIC STAGE TOKENS

## ABSTRACT

In recent years, using predefined agentic workflows to guide large language models (LLMs) for literature classification and review has become a research focus. However, writing research introductions is more challenging. It requires rigorous logic, coherent structure, and abstract summarization. Existing workflows often suffer from long reasoning chains, error accumulation, and reduced textual coherence. To address these limitations, we propose eliminating external agentic workflows. Instead, we directly parameterize their logical structure into the LLM. This allows the generation of a complete introduction in a single inference. To this end, we introduce the Stage Token for Introduction Generation (STIG). STIG converts the multiple stages of the original workflow into explicit stage signals. These signals guide the model to follow different logical roles and functions during generation. Through instruction tuning, the model learns the mapping between stage tokens and text functions. It also learns the logical order and transition patterns between stages, encoding this knowledge into the model parameters. Experimental results show that STIG can generate multi-stage text in a single inference. It does not require explicit workflow calls. STIG outperforms traditional agentic workflows and other baselines on metrics of semantic similarity and sentence-level structural rationality. The code is provided in the Supplementary Materials.

## 1 INTRODUCTION

In recent years, large language models (LLMs) and LLM agents have become essential tools throughout the entire scientific research lifecycle (Zhang et al., 2025b; Lu et al., 2024). Early studies have shown that they can accelerate scientific discovery (Garikaparthi et al., 2025; Li et al., 2025; Langley, 2024; Wang et al., 2023), generate innovative research hypotheses (Yang et al., 2024b), and even participate in experimental design and execution (Zhao et al., 2025; Boiko et al., 2023; Huang et al., 2024). Now, LLMs have been integrated into interactive research agents, enabling end-to-end support from academic database retrieval to iterative manuscript refinement.

However, current LLM-based approaches for academic writing still face critical limitations that hinder broad adoption. Many existing methods rely on agentic workflows (Liu et al., 2025), which divide the writing process into discrete stages, such as background composition and problem articulation. Specialized agents are assigned to each stage. These workflows depend heavily on carefully handcrafted designs for agent roles, invocation sequences, and fallback strategies. Any change in task granularity or domain conventions may require costly workflow reconfiguration. These methods also rely on multi-turn iterative interactions and opaque API calls. This increases token usage, slows inference, and adds substantial computational overhead. A typical generation pipeline is illustrated in the upper half of Figure 1. Prior LLM-based academic writing research, exemplified by AutoSurvey (Wang et al., 2024b) and SURVEYFORGE (Yan et al., 2025), has mostly focused on literature reviews. These methods emphasize document classification and summariza-

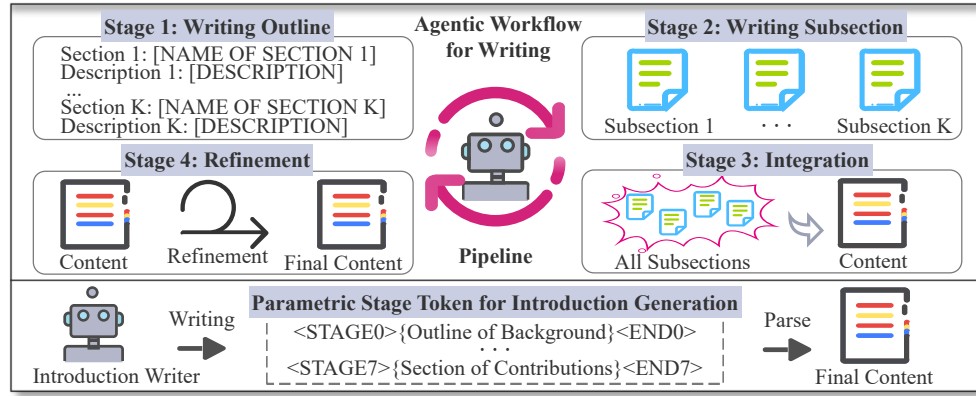

Figure 1: Comparison of Agentic Workflow and STIG in writing. Agentic workflows rely on multi-turn interactions. They complete writing step by step, including outline generation, subsection drafting, integration, and refinement. In contrast, STIG generates the content in a single inference. The final output is obtained by parsing the generated text according to stage tokens.

tion. However, generating specific manuscript sections, such as the Introduction, has been largely neglected. The Introduction is a core section that integrates research background, objectives, and contributions into a coherent and logically rigorous narrative. Its quality directly affects the clarity, logical flow, and scholarly impact of a paper. Yet existing workflow-based LLM agents often omit or fabricate critical content, such as experimental results or baseline comparisons, which undermines the paper's factual accuracy and persuasiveness.

To address cascading errors, structural drift, and computational redundancy caused by brittle, manually designed agentic workflows, we propose the Stage Token for Introduction Generation (STIG) model. STIG compresses multi-stage writing logic into a single inference using parametric stage tokens. This eliminates the need for manually orchestrated workflows. The model converts the multiple stages of the original workflow into explicit stage signals. These signals guide the model to follow different logical roles and functions during generation. Through instruction tuning, the model learns the mapping between stage tokens and text functions. It also learns the logical order and transition patterns between stages. This knowledge is directly encoded into the model parameters. We construct a high-quality training dataset by parsing over 2,600 scientific papers. Each sample is split into eight sequential subtasks across four core subsections: Background, Problem Statement, Methodological Overview with Results, and Research Contributions. Each subsection is further divided into Outline Generation and Content Drafting. All samples are annotated with stage tokens, such as $\langle$STAGE0$\rangle$ for *Background outlines* and $\langle$STAGE1$\rangle$ for *Background content*. This explicitly encodes the temporal logic of introduction writing. Using this dataset, we fine-tuned open-source LLMs to generate structured and academically rigorous introductions in a single inference. The process is illustrated in the lower half of Figure 1.

We conduct experiments on 1,176 papers from the ACL 2025 Main Conference, comparing STIG with one-shot methods such as Pure Prompt and prevalent agentic-workflow baselines. We evaluate the generated introductions at both the overall semantic level and sentence-level structural rationality. Experimental results show STIG model eliminates agentic workflow dependencies and achieves a computational efficiency of up to 3.3 times that of agentic workflows, while generated introductions also outperform those from agentic workflows and other baselines in terms of semantic similarity and structural rationality metrics.

This work advances academic introduction writing through three key contributions. (1) We introduce STIG model, which eliminates agentic workflow by integrating parametric stage. (2) We decompose introduction

writing into multiple stages and construct the corresponding training data, integrating stage tokens into both model training and inference. During inference, stage-wise generation is accomplished by conditioning on these stage tokens. Additionally, we construct a customized dataset tailored for training and testing introduction generation, derived from over 3,800 ACL main conference papers. (3) Experimental results demonstrate STIG model outperforms agentic workflows on composite metrics like semantic similarity, structural rationality, and content coverage, producing introductions that better conform to academic norms.

## 2 RELATED WORK

### 2.1 LLM AGENTIC WORKFLOW

Agentic workflows coordinate multiple LLM agents to address complex tasks by decomposing them into several modulars. In scientific research, these workflows emulate collaborative teams, dividing the academic writing process into stages, including idea generation (Su et al., 2025; Wu et al., 2023; Baek et al., 2025), literature review (Zimmermann et al., 2024; Agarwal et al., 2024), experimental design (Ye et al., 2025), and results analysis (Schmidgall et al., 2025). Existing studies focus on developing communication protocols, reflection strategies to enhance agent self-awareness, and adaptive autonomous multi-agent systems that adjust to task variations (Zhang et al., 2025a; Hu et al., 2025a). However, these workflows require meticulous design of agent interactions and task assignments, significantly increasing complexity and limiting scalability for practical applications (Liu et al., 2025). Additionally, reliance on opaque API calls restricts access to model parameters, while multi-turn interactions incur high token costs and latency. To this end, our STIG model employs stage tokens to enable end-to-end generation of introductions in a single inference step, reducing computational costs.

### 2.2 LLM AGENTS FOR SCIENTIFIC PAPER WRITING

LLM agents support academic writing tasks, including citation generation, literature synthesis, and section drafting. For citation generation, multimodal networks generate intent-aware summaries, while graph-based clustering organizes related studies (Ge et al., 2021; Wang et al., 2021; Hu et al., 2025b). In writing tasks, LLM agents apply task decomposition and retrieval-augmented generation to automate literature reviews through multi-stage pipelines of retrieval, outlining, drafting, and refinement, achieving high citation recall (Wang et al., 2024b). Other methods improve coherence using outline heuristics and memory-driven refinement (Yan et al., 2025). Multi-agent systems with iterative reinforcement learning support drafting and simulated analysis, often relying on fabricated experimental data (Weng et al., 2025). However, prior research has narrowly concentrated on literature review generation. Compared with the encyclopedic coverage sought by surveys, introduction writing poses a sharper rhetorical challenge because it must simultaneously map the scholarly background, pinpoint the research gap, and foreshadow the empirical contribution. The absence of any single argumentative link disrupts textual coherence and precipitates the omission of critical evidentiary elements such as experimental outcomes (Garg et al., 2025). Our STIG model introduces a curated dataset and stage tokens to generate structured, authentic introductions for scientific papers.

## 3 STAGE TOKEN FOR INTRODUCTION GENERATION

### 3.1 GENERATION WITH STAGE TOKEN

STIG model generates introductions for scientific papers using core materials from a completed paper, denoted as $\mathcal{M} = \{\mathcal{T}, \mathcal{A}, \mathcal{F}, \mathcal{TA}, \mathcal{R}\}$, where $\mathcal{T}$ is the title, $\mathcal{A}$ is the abstract, $\mathcal{F}$ is the descriptions of the paper's figures, $\mathcal{TA}$ represents descriptions and table contents of the paper's tables, and $\mathcal{R}$ is the abstracts of baseline references, providing context. $\mathcal{T}$ and $\mathcal{A}$ provide the research theme and core logic of the study while

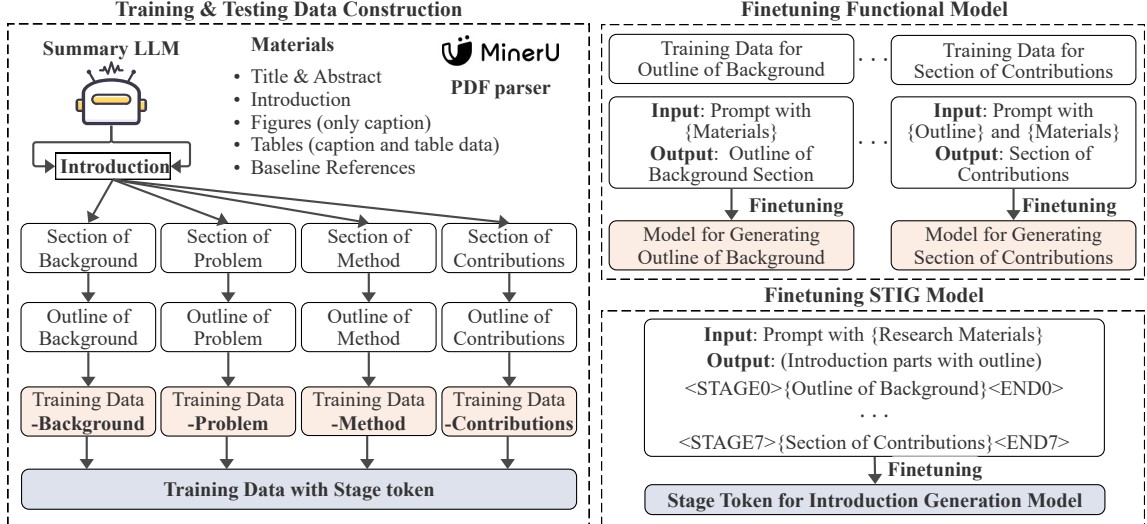

Figure 2: **Overview of the STIG Model Architecture**. STIG integrates eight stage-token pairs that guide the generation of four subsections, across outline and content phases. We highlight the flow from core input materials to structured output, emphasizing the single inference mechanism and the role of parametric stage tokens in enhancing logical coherence. The top-right corner illustrates the stage writing finetuning procedure, which constitutes a component of our ablation study in section 5.2.

$\mathcal{F}$, $\mathcal{TA}$, and $\mathcal{R}$ collectively serve as content support for introduction generation. Traditional outline-based methods first produce an outline $\mathcal{O} = \{o_1, o_2, \ldots, o_n\}$ leveraging $\mathcal{M}$ as input where $\mathcal{O} = \text{LLM}(\mathcal{M})$. Then, paragraphs are generated for each outline point and merged into the final introduction $\mathcal{I}$:

$$\mathcal{I} = \text{Merge}(s_0, s_1, \cdots, s_n), \tag{1}$$

where $s_i = \text{LLM}(o_i)$ for $i \in [0, n]$. Specifically, an LLM agent generates a dedicated paragraph $s_i$ for each individual outline point $o_i$. Merge$(\cdot)$ function consolidates all generated paragraphs $\{s_0, s_1, \cdots, s_n\}$ into a cohesive introduction $\mathcal{I}$. This multi-step process introduces inefficiencies due to sequential calls and potential inconsistency.

The STIG model integrates an end-to-end generation process for Introductions, using core materials $\mathcal{M} = \{\mathcal{T}, \mathcal{A}, \mathcal{F}, \mathcal{TA}, \mathcal{R}\}$ to guide structured output. To enforce a fixed generation order, it employs eight stage-token pairs, from $\langle\text{STAGE0}\rangle$-$\langle\text{END0}\rangle$ to $\langle\text{STAGE7}\rangle$-$\langle\text{END7}\rangle$, corresponding to outline and content tasks for four subsections: Background, Problem and Limitations of Existing Methods, Brief Method Overview and Summary of Main Results, and Contributions. For instance, $\langle\text{STAGE0}\rangle$-$\langle\text{END0}\rangle$ denotes the Background outline, and $\langle\text{STAGE1}\rangle$-$\langle\text{END1}\rangle$ its content, decomposing the process into eight sequential subtasks to provide clear stage signals and support the logic of outline-to-content and modular generation.

This end-to-end approach mitigates inefficiencies from multi-agent workflows by generating the entire Introduction in a single inference step. The token sequence for stage $k + 1$, $\mathcal{S}k+1 = \{\langle\text{STAGE}k + 1\rangle, s_{k+1,1}, s_{k+1,2}, \ldots, \langle\text{END}k + 1\rangle\}$, is predicted based on the concatenated prior stages $\mathcal{S}_{<k+1} = \mathcal{S}_0 \oplus \mathcal{S}_1 \oplus \cdots \oplus \mathcal{S}k$, where $\mathcal{S}i = \{\langle\text{STAGE}i\rangle, s_{i1}, s_{i2}, \ldots, \langle\text{END}i\rangle\}$ includes stage tokens and content. The joint probability of the introduction is:

$$P(\mathcal{S}_0 \oplus \mathcal{S}_1 \oplus \cdots \oplus \mathcal{S}_n) = \prod_{t=1}^{T} P(S_t \mid S_{<t}). \tag{2}$$

The probability of generating tokens for the $(k+1)$-th stage is:

$$p_\theta(\mathcal{S}_{k+1} \mid \mathcal{S}_{<k+1}) = \prod_{t \in \mathcal{T}_{k+1}} p_\theta(y_t \mid \mathcal{S}_{<k+1} \oplus \mathcal{Y}_{<t}). \tag{3}$$

Here, $\mathcal{T}_{k+1}$ denotes the set of position indices of all tokens in the $(k+1)$-th stage (*e.g.*, $\mathcal{T}_1$ corresponds to the positions of $\langle \text{STAGE1} \rangle$, Background content tokens, and $\langle \text{END1} \rangle$). $\mathcal{S}_{<k+1} \oplus \mathcal{Y}_{<t}$ represents the concatenation of the complete sequence of the previous $k$ stages and the preceding tokens $\mathcal{Y}_{<t}$ of the $(k+1)$-th stage. This ensures that when the model generates the current token, it can simultaneously refer to the logic of all previous stages and the content already generated in the current stage.

During training process, the stage association logic of from previous outline to current content and from current section to next section is internalized into the model parameters $\theta$. The training model can automatically complete the prediction and generation of previous from stage to next stage based on Equation 2 without manual intervention, realizing end-to-end Introduction generation.

## 3.2 DATA CONSTRUCTION

To train STIG model effectively, we constructed a dataset through a two-step process of batch collection and structured conversion. First, we collected long papers from ACL 2021–2025 Main Conferences via the official open interface of the ACL Anthology digital library [1]. Second, we employed the MinerU tool (Wang et al., 2024a) to perform structured parsing on these PDFs, converting contents into Markdown and JSON formats. We extract key materials including title, abstract, introduction, figures, and tables, providing data foundation for model input preparation. In total, we obtained over 3,800 papers, with 1,176 of them (ACL 2025) utilized as test data. As a supplement, we employ GPT-4o (Achiam et al., 2023) to extract citation identifiers (*e.g.*, Smith et al., 2023) from experimental sections and match them with reference lists to retrieve metadata and obtain full abstracts via the Semantic Scholar API, ultimately creating an auxiliary dataset of baseline references.

For STIG's phased generation, a triple framework of core materials, subsection outlines, and subsection content directs the annotation process. We employ GPT-4o to annotate introduction sections, targeting four subsections. Each subsection received two types of annotations: outline annotations extracted logical key points in list form (*e.g.*, current research status and bottlenecks for the Background subsection), and content annotations merged original sentences into coherent paragraphs aligned with the outlines, ensuring one-to-one correspondence between them. Annotation results are stored in with each sample including subsection outline and subsection content, as detailed in Figure 2, details are shown in Appendix B. Based on these annotations, eight specialized training groups are constructed, corresponding to outline and content tasks for each subsection, providing precise supervision for STIG's phased learning.

## 3.3 TRAINING AND INFERENCE

All models are trained on $8 \times$ A800 GPUs using the LLaMA-Factory framework (Zheng et al., 2024) with ZeRO3 optimization (Rajbhandari et al., 2020) to handle memory constraints efficiently during the finetuning process. We maximize context length to 8K tokens while training and add 8 groups of special tokens.

During the inference phase, the trained model generates content in a structured manner, producing text containing Stage tokens. It then removes the outline sections, extracts the main content sections, and finally concatenates the four parsed sections to form the final Introduction.

---

[1] https://aclanthology.org/

## 4 EXPERIMENTAL SETTINGS

### 4.1 BACKBONE LLMS AND BASELINES

We employ two widely used open-source LLMs as backbones: Qwen2.5-7B-Instruct (Yang et al., 2024a) and Llama3.1-8B-Instruct (Dubey et al., 2024) and we compared our proposed method with the following baselines:

**Pure Prompt / ELABORATE Prompt**: A baseline using only prompt engineering without explicit outline guidance, where the model generates the introduction directly based on input materials. ELABORATE prompt (Garg et al., 2025) enforces a strict four-paragraph structure (each 100–150 words), detailing context, gaps, contributions, and impact.

**GPT**: Generated directly by GPT-4o, with two settings: pure prompt and elaborate prompt.

**Outline Writing**: A two-stage approach where the model first generates an outline for the introduction and then expands it into full content sequentially.

**AutoSurvey (Wang et al., 2024b)**: An advanced outline-based method that incorporates survey-style structured prompting to guide the outline generation process, enhancing the coverage of existing literature.

**STIG (Our method)**: Our proposed STIG model, which eliminates agentic workflows and integrates stage tokens into the sequence generation process after SFT to learn the phased writing logic.

### 4.2 MULTI-DIMENSIONAL EVALUATION METRICS

To comprehensively assess generated introductions, we adopt multi-dimensional evaluation metrics covering semantic similarity, structural rationality, content coverage, narrative quality, and hard constraints.

**Semantic Similarity (SS)** evaluates the consistency of meaning between the generated introduction and the original introduction. It measures the degree to which the generated introduction is semantically accurate. We adopt BERTScore (Zhang* et al., 2020) as the specific metric for this purpose.

**Structural Rationality (SR)** verifies adherence to the introduction framework (Background, Problem and Limitations, Method Overview, Research Contributions) by checking subsection boundaries. It quantifies content confusion avoidance, labeling sentences and computing the error rate from misclassified sentences ($C_{\mathrm{mis}}$) over total sentences ($C_{\mathrm{total}}$):

$$\text{Structural Rationality} = 1 - \frac{C_{\mathrm{mis}}}{C_{\mathrm{total}}}. \tag{4}$$

**Content Coverage (CC)** measures the extent to which a generated introduction captures key contents from the original, combining content completeness and structural rationality. It calculates sentence-level SBERT similarity ($s_j$) between each generated subsection sentence and the corresponding original subsection, weighted by a rationality label ($r_j$: 1 for correct classification, 0 for incorrect). The base score averages weighted similarities, adjusted by a missing coefficient ($k$: 1 for no missing subsections, 0.75 for one):

$$\text{CC} = \left( \frac{1}{m} \sum_{j=1}^{m} (s_j \times r_j) \right) \times k, \tag{5}$$

where $m$ denotes the total number of sentences.

**Narrative Quality (NQ)** evaluates fluency and readability. We employ Perplexity (PPL), computed as average negative log-likelihood per token under GPT-2 (Radford et al., 2019) as the metric with lower values

Table 1: **Quantitative results of STIG and baselines**. All methods are tested on 1,176 main conference papers from ACL 2025. Our method outperforms the baselines overall, particularly in terms of structural rationality and content coverage. Additionally, as Llama3.1-8B-Instruct is unable to produce the outline format required by AutoSurvey, the corresponding experiments could not be conducted. Underlined text indicates poor readability.

| LLM | Methods | SS | SR | CC | NQ | QC |
|---|---|---|---|---|---|---|
| GPT-4o | GPT | **0.973** | 0.779 | 0.399 | 28.233 | **1.00** |
| | GPT (Elaborate) | 0.971 | **0.903** | **0.496** | 31.906 | 1.00 |
| Qwen2.5-7B-Instruct | Pure Prompt | 0.975 | 0.749 | 0.394 | 25.426 | 0.95 |
| | ELABORATE Prompt | 0.972 | 0.722 | 0.327 | 28.461 | 0.97 |
| | Outline Writing | 0.972 | 0.706 | 0.357 | 28.613 | 0.99 |
| | AutoSurvey | 0.966 | 0.658 | 0.333 | **18.084** | 0.92 |
| | STIG (Ours) | **0.977** | **0.832** | **0.442** | 24.810 | **1.00** |
| Llama3.1-8B-Instruct | Pure Prompt | 0.975 | 0.772 | 0.427 | **14.843** | 1.00 |
| | ELABORATE Prompt | 0.980 | 0.800 | 0.447 | 19.981 | 1.00 |
| | Outline Writing | 0.958 | 0.759 | 0.404 | 26.328 | 1.00 |
| | AutoSurvey | — | — | — | — | — |
| | STIG (Ours) | **0.978** | **0.836** | **0.472** | 20.717 | **1.00** |

(below 25 for strong readability) signifying better quality. For sequence $T = [t_1, t_2, ..., t_n]$:

$$\text{PPL}(T) = \sqrt[n]{\prod_{i=1}^{n} \frac{1}{P(t_i \mid t_1, ..., t_{i-1})}}, \qquad (6)$$

where $P(t_i \mid t_1, ..., t_{i-1})$ is the conditional probability of token $t_i$. To avoid numerical underflow,

**Hard Constraints** assess instruction following ability, focusing on Quotation Constraint (QC). Explicit instructions exclude citations, and non-compliance occurs if markers (*e.g.*, *Smith et al., 2023, [12]*) appear.

## 5 EXPERIMENTS AND ANALYSIS

### 5.1 QUANTITATIVE RESULTS

We conduct comparative analysis between baselines and our STIG model, with results presented in Table 1. GPT-4o with elaborate prompt achieves the highest SR and CC score. The experimental findings substantiate STIG's efficacy in elevating the quality of generated introductions, with a particular emphasis on structural rationality, a critical metric reflecting adherence to the academic introduction writing framework. This pronounced structural coherence underscores the strategic advantage of STIG's stage-token training, which meticulously delineates module boundaries, mitigates content confusion, and ensures logical progression through SFT. STIG outperforms baselines by up to 17.4% in structure rationality (*e.g.*, 0.832 vs. 0.658 against AutoSurvey on Qwen2.5-7B-Instruct), leveraging parametric stage tokens to encode phased generation logic. Complementing this, STIG achieves superior content coverage and semantic similarity, reflecting robust content fidelity and completeness. In contrast, relying on agentic workflows, Outline Writing and AutoSurvey exhibit structural deficiencies due to isolated module generation followed by post-hoc integration, amplifying cascading errors and yielding performance inferior to the simpler Pure Prompt baseline,

> Managing intricate tasks that necessitate iterative dialogue and feedback poses significant challenges for large language models (LLMs). As these models increasingly engage with complex environments, their effectiveness hinges on the ability to efficiently incorporate feedback into successive interactions. Traditional methods such as sequential revision and parallel sampling struggle with length generalization and lack the ability to self-reflect, leading to suboptimal performance. Moreover, naive retry mechanisms fail to leverage prior knowledge, further hindering progress. To address these limitations, we introduce FTTT (Feedback-Enabled Test-Time Training), a paradigm that formulates feedback utilization as an optimization problem at test time. We also propose OpTune ⋯.
>
> Addressing complex tasks that require iterative refinement and continuous feedback presents significant challenges for current methods. Existing schemes such as Revision (Snell et al., 2024), ⋯ leading to suboptimal performance. Beam Search, despite its efficiency, also falters in length generalization. In contrast, our proposed approach, FTTT, ⋯ overcome these limitations through a robust optimization framework at test time. Figures 1 and Table 1 illustrate⋯.
>
> Next, we introduce Fine-Tuning Through Testing-Time Optimization (FTTT), a novel framework that formulates feedback ⋯
>
> ● Background            ● Problem            ● Method

Figure 3: **The introduction generated by AutoSurvey, with each sentence annotated to correspond to one of the four subsections**. The introduction generated by AutoSurvey exhibits a highly irrational structure, with excessive and repetitive emphasis on the method. Detailed introduction is shown in D.2.

Table 2: **Ablation results**. We test the effectiveness of finetuning and stage writing. Among them, finetuning brings stable performance improvements, while stage writing needs to be combined with finetuning to achieve more effective performance enhancements due to deviations in the agentic workflow.

| LLM | Methods | SS | SR | CC | NQ | QC |
|---|---|---|---|---|---|---|
| Qwen2.5-7B-Instruct | FT w/o Stage Writing | **0.980** | 0.797 | 0.433 | 26.350 | 1.00 |
| | Stage Writing w/o FT | 0.971 | 0.682 | 0.390 | **20.341** | 0.96 |
| | Stage Writing FT | 0.978 | 0.800 | 0.430 | 38.174 | 1.00 |
| | STIG (Ours) | 0.977 | **0.832** | **0.442** | 24.810 | **1.00** |
| Llama3.1-8B-Instruct | SFT w/o Stage Writing | 0.980 | 0.790 | 0.440 | 20.054 | 1.00 |
| | Stage Writing w/o FT | 0.968 | 0.819 | 0.450 | **13.864** | 0.98 |
| | Stage Writing FT | **0.980** | **0.842** | **0.495** | 27.183 | 1.00 |
| | STIG (Ours) | 0.978 | 0.836 | 0.472 | 20.717 | **1.00** |

as evidenced by its inconsistent structural rationality and content coverage scores. As shown in Figure 3, introduction generated by AutoSurvey exhibits extreme academic irregularity and weak logical coherence.

## 5.2 ABLATION STUDIES

The ablation study, as detailed in Table 2, substantiates the pivotal roles of stage writing and finetuning in enhancing the efficacy of the STIG model.

**FT w/o Stage Writing**: This approach involves finetuning the model on the annotated dataset without incorporating outline and stage tokens or phased writing logic. Pure finetuning significantly elevates semantic similarity, yet its performance in structural rationality remains constrained, suggesting that while this approach optimizes semantic fidelity, it inadequately ensures the structural coherence of generated introductions.

**Stage Writing w/o FT** employs agentic workflow to perform stage-based writing (from outline to content) across the four subsections, relying on agent interactions without SFT. The stage writing methodology em-

Table 3: **Performance of STIG variants with different stage granularity.** Eight stages refers to outline followed by content drafting for each of the four modules while four stages refers to direct content drafting without outlines.

| Methods | SS | SR | CC | NQ | QC |
|---|---|---|---|---|---|
| STIG (Four Stages) | 0.976 | 0.808 | 0.425 | 24.189 | 1.00 |
| STIG (Eight Stages) | 0.977 | 0.832 | 0.442 | 24.810 | 1.00 |

beds a deliberate structural design intent, but unmitigated errors propagate through the agentic workflow and produce suboptimal structural rationality scores.

**Stage Writing FT** also employs the same multi-agent system to perform stage-based writing across the four subsections and trains all the agents with LoRA (Hu et al., 2022) (LoRA reduce memory usage and enable easy switching). Differently, it integrates task-specific finetuning into the stage writing framework, which effectively mitigate the inherent error propagation of agentic workflows, yielding performance levels approaching that of STIG. Details are shown in Figure 2.

STIG model combining supervised finetuning with the internalization of stage writing into parametric stage tokens achieves superior structural rationality while closely aligning semantic content with the original introduction. This holistic enhancement is quantitatively supported by STIG's elevated content coverage score, surpassing that of competing models.

### 5.3 IMPACT OF STAGE GRANULARITY

We investigate the effect of stage granularity on generation quality by comparing the full STIG model, which employs eight stages against a variant with four stages. Results on Qwen2.5-Instruct-7B are shown in Table 3. The eight-stage configuration outperforms the four-stage variant on all metrics because explicit outline guidance strengthens logical organization and reduces subsection confusion. The finer granularity achieved by separating outline and content phases enhances structural coherence and content fidelity without sacrificing performance.

### 5.4 GENERALIZATION STUDY

To evaluate generalization capability of STIG to other domains, we conduct experiments on a dataset comprising 102 CVPR papers on Qwen2.5-Instruct-7B, whereas STIG model is originally trained on ACL data. The results are shown in Table 10 in Appendix D.4. STIG outperforms baselines except GPT with elaborate prompt, demonstrating robust structure alignment and content capture despite the domain shift. Notably, Stage Writing FT underperforms STIG, as it lacks module linkage during training, resulting in performance degradation under domain shift while STIG enhances transferability through integrated stage training.

### 5.5 HUMAN EVALUATION AND CASE STUDY

We conduct a human evaluation on 50 ACL papers to validate model performance. There evaluators are given each paper's title and abstract as context, and introductions were generated using seven methods. To eliminate bias, the seven introductions per paper were shuffled. Expert researchers discuss and rank them from 1 (best) to 7 based on coherence, completeness, and adherence to academic standards. Table 4 summarizes the ranking distribution and average ranks, showing that STIG model attains the highest average rank, indicating superior perceived quality, followed by FT w/o Stage Writing. However, GPT-generated

Table 4: **Human evaluation ranking distribution and average ranks across seven methods.** STIG model achieves the highest ranking, followed by FT w/o Stage Writing while AutoSurvey performed the worst.

| Method | Rank 1 | 2 | 3 | 4 | 5 | 6 | Rank 7 | Avg Rank |
|---|---|---|---|---|---|---|---|---|
| GPT | 2 | 13 | 15 | 7 | 4 | 5 | 4 | 3.58 |
| GPT (ELABORATE) | 4 | 3 | 14 | 13 | 11 | 4 | 1 | 3.80 |
| ELABORATE Prompt | 0 | 2 | 9 | 11 | 10 | 10 | 8 | 4.82 |
| AutoSurvey | 0 | 0 | 3 | 2 | 11 | 14 | 20 | 5.92 |
| FT w/o Stage Writing | 18 | 17 | 3 | 5 | 2 | 3 | 2 | 2.46 |
| Stage Writing w/o FT | 0 | 2 | 5 | 8 | 9 | 14 | 12 | 5.28 |
| STIG (Ours) | 26 | 13 | 1 | 4 | 3 | 0 | 3 | **2.14** |

introductions often contain excessive claims of achievements and fail to align with academic conventions and without aligning with academic standards. Table 13 shows the detailed ranking information.

## 5.6 EFFICIENCY ANALYSIS

We investigate efficiency of various methods using Qwen as backbone, quantifying total token consumption and the number of effectively generated tokens, as depicted in Figure 4. STIG model exhibits a marked efficiency advantage, consuming the fewest total tokens, attributable to its streamlined prompt de-

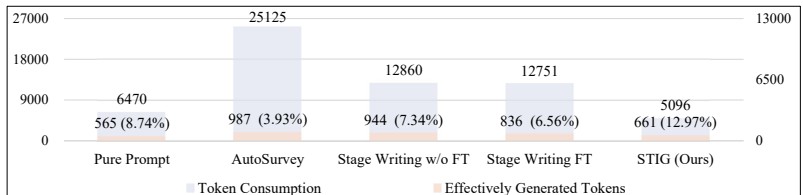

Figure 4: **Statistics on token consumption and effectively generated tokens**. Our method demonstrates the highest token usage efficiency.

sign and end-to-end generation approach, achieving an effectiveness rate 3.3 times that of AutoSurvey. Compared to Stage Writing FT, which undergoes similar finetuning and delivers comparable performance, STIG model demonstrates double the efficiency, as the latter incurs greater token expenditure due to overhead of multi-agent interactions. This reduction in token usage, particularly in total token count, underscores STIG model's capability to simplify generation process and minimize redundant computations.

## 6 CONCLUSION AND LIMITATIONS

**Conclusion.** This paper introduces the STIG model, which eliminites external agentic workflows. By parameterizing stage token, it internalizes the logical structure of the agentic workflow into the LLM itself, enabling single generation that avoids cascaded errors and inefficiencies of agentic workflows for introduction writing of scientific papers. Experimental evaluations across diverse model backbones demonstrate STIG's superiority, exhibiting enhanced structural rationality, semantic fidelity, and content coverage while maintaining narrative coherence and adherence to academic norms. These outcomes affirm the STIG model's capacity to streamline phased generation and reduce computational demands.

**Limitations.** This study acknowledges some inherent limitations of the STIG model. Firstly, the training of the model relies on datasets from specific domains and specific structures, which may limit its generalization to other academic fields or manuscript types beyond computer science conference papers. Furthermore, the model's reliance on annotated data assumes high-quality annotations, and any inconsistency may affect performance.

ETHICAL STATEMENT

In this study, we only employ LLMs as an exploratory tool to generate the introduction section of conference papers in the field of computer science. The core objective is to evaluate their auxiliary potential in academic research, not to advocate for the unsupervised use of LLMs in academic writing scenarios. While LLMs can produce initial drafts of sections that appear logically coherent and format-compliant with academic standards, they are prone to significant issues such as factual errors, including misleading representations of fundamental concepts in the field and inaccuracies in key technical details.

This study emphasizes that all academic content generated by LLMs must undergo sentence-by-sentence verification, factual validation, and targeted revision by researchers to ensure the content's accuracy, rigor, and academic integrity. This process further confirms that throughout the entire academic writing work-flow, the role of human researchers in providing professional judgment, knowledge verification, and quality control is irreplaceable, serving as a core link in safeguarding the credibility of academic outcomes.

REPRODUCIBILITY STATEMENT

This study ensures the full reproducibility of the Stage Token for Introduction Generation (STIG) model by providing open access to supplementary materials and documentation.

The source code, including scripts for model generation (main experiments and ablation studies) and eval-uation (semantic similarity, structural rationality, content coverage, narrative quality and hard constraints), is presented in the supplementary materials. Detailed model configuration information is documented in the experimental setup, encompassing the Qwen2.5-7B-Instruct and Llama3.1-8B-Instruct backbones. LLMs are trained employing the LLaMA-Factory framework and ZeRO3 optimization. For evaluating structural rationality, this study employs the Qwen2.5-32B-Instruct model.

The ACL dataset used in this study for training and testing contains annotated introduction sections of papers. It is available for research purposes, though its usage must comply with relevant licensing terms. Potential challenges include hardware dependencies on high-performance GPUs and the need for consistent annotation quality, which may affect replication across diverse environments.

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

## A  THE USE OF LARGE LANGUAGE MODELS (LLMs)

In this study, LLMs are used exclusively for grammar correction and text polishing to improve readability. They did not contribute to research ideation, content generation, or any substantive aspects of the work. We (the authors) bear full responsibility for all contents, ensuring originality and accuracy.

## B  ANNOTATION OF TRAINING AND TESTING DATA

The following is the prompt used in this paper to extract the outline and content of the four subsections via GPT-4o. A metadata sample immediately follows.

```
Prompt for Extracting

Please break down the introduction section of the following academic
    paper into a structured outline format for academic discussion
    purposes.
The content should be divided into the following four sections,
    extracting key points for each:

1. Background: Basic background and significance of the research field
   - Number of points: 2-4
2. Problem and Limitations of Existing Methods: Current issues,
   challenges, and limitations of existing methods
   - Number of points: 2-6
3. Brief Method Overview and Summary of Main Results: Overview of the
    proposed method, main experimental results, and findings
   - Number of points: 4-8
4. Our Contributions: Main contributions and innovations of the paper
   - Number of points: 2-3

Please output in the following JSON format, including outline points and
    paragraphs classified by section:

{
    ``sections'': {
        ``Background'': ``Combine all paragraphs and sentences belonging
            to the background section'',
        ``Problem and Limitations of Existing Methods'': ``Combine all
            paragraphs and sentences belonging to the problems and
            limitations section'',
        ``Brief Method Overview and Summary of Main Results'': ``Combine
            all paragraphs and sentences belonging to the method overview
            and main results sections'',
        ``Our Contributions'': ``Combine all paragraphs and sentences
            belonging to the contributions section''
    },
    ``outline'': {
        ``Background'': [``Point 1'', ``Point 2'', ...],
        ``Problem and Limitations of Existing Methods'': [``Point 1'',
            ``Point 2'', ...],
        ``Brief Method Overview and Summary of Main Results'': [``Point
            1'', ``Point 2'', ...],
```

```
            ``Our Contributions'': [``Point 1'', ``Point 2'', ...]
        },
    }

    Introduction content:
    {text}

    Notes:
    1. Analyze the content coherently and categorize it into the appropriate
        sections, strictly controlling the number of points for each section.
    2. When assigning sections, ensure continuity; for example, Background
        must be at the beginning of the article, and if there is an Our
        Contributions section, it must be at the end. In some paragraphs, the
        first part might belong to section 1, and the latter part to section
        2. There should be no section 1, section 2, section 1 sequences.
    3. Some papers may not have a section similar to Our Contributions; if
        so, generate an empty Our Contributions field.
    4. First, divide the sections, then perform an outline analysis to
        identify key points.
    5. Do not use demonstrative pronouns like ``this'' or ``the model'' in
        the key points; use specific names if available.
```

**Metadata of Training and Testing data**

```
{
    ``sections'': {
        ``Background'': ``Text embeddings, essential language features,
            are foundations of semantic textual similarity (STS) tasks,
            which quantify how similar two text pieces are in semantics.
            They broadly benefit downstream tasks, such as information
            retrieval and clustering, and are particularly helpful in many
            recent LLMs-based applications; e.g., many RAG tasks employ
            text embeddings for retrieval.'',
        ``Problem and Limitations of Existing Methods'': ``The existing
            STS training commonly involves optimizing cosine functions -
            the learning objective to indicate the similarity of pairwise
            text embeddings. However, the cosine has saturation zones,
            resulting in gradient vanishing in optimization regardless of
            the network depth. The gradient will be close to zero for
            embedding pairs falling in the saturation zone, preventing
            parameters from updating in backpropagation. Because embedding
            pairs in saturation zones are nearly aligned or antialigned, it
            hinders text embedding models from discerning subtle, implicit
            differences that appear similar yet are actually dissimilar in
            semantics. Such pairs commonly appear in STS training data from
            Natural Language Inference (NLI) datasets, such as the
            Multi-Genre NLI (MNLI) and the Stanford NLI (SNLI). They
            typically include three labels of entailment, neutral, and
            contradict; pairs in saturation zones may render obscure
            cross-label boundaries. To illustrate this point, Figure 1
            shows an example from the SNLI dataset. The ``neutral'' pair
```

```
            shows a high appearance similarity (with many shared words)
            instead of semantically similar. The similar appearance
            similarity results in them falling into cosine's saturation
            zones, causing vanishing gradients during optimization.
            Consequently, the model mistakenly considers their relations as
            ``entailment'' instead of their correct label ``neutral.''''',
    ``Brief Method Overview and Summary of Main Results'': ``Viewing
            these concerns, we aim to tackle the negative effects of the
            cosine's saturation zones in embeddings and propose a novel
            Angle-optimized Embedding (AoE) model for STS. It decomposes an
            embedding into real and imaginary components through complex
            division, aiming to employ the real component for reflecting
            appearance differences and the imaginary component for subtle
            differences. It allows AoE to involve the optimization of the
            angle difference to understand subtle differences in text pairs
            for similarity learning. To the best of our knowledge, we are
            the first to explore the negative effects of cosine's
            saturation zones and optimize angle differences through
            division in complex space for text embedding learning. In the
            STS experimental setup, we observed that most existing STS
            benchmarks focus on evaluating models on short texts.
            Unfortunately, limited datasets are available to evaluate the
            STS performance on long texts. However, long texts are
            prevalent in real-world applications such as financial and
            legal documents. To tackle this challenge, we present a
            high-quality long-text STS dataset collected from GitHub Issues
            with roughly 22K samples. It allows for a more comprehensive
            evaluation of STS performance with long texts. We first
            experimented with short- and long-text STS datasets in the
            standard and in-domain STS tasks, where AoE outperforms
            non-trivial baselines in varying embedding backbones. Then, AoE
            shows consistently superior results in facilitating various
            downstream tasks, indicating its benefits in diverse scenarios.
            In particular, AoE achieves SOTA results on the Massive Text
            Embedding Benchmark (MTEB) at the same model scale. Next, an
            ablation study indicates that all modules positively contribute
            to AoE. Finally, we further discuss how AoE learns better
            embeddings in cosine saturation zones.'',
    ``Our Contributions'': ``In summary, our contributions are as
            follows: We investigate the effects of cosine saturation zones
            for STS and optimize angle differences in complex space for
            improving text embedding. We extend the existing STS benchmark
            with a new long-text dataset from Github Issues to allow more
            comprehensive STS empirical studies. We present extensive
            experiments demonstrating that AoE effectively handles cosine
            saturation zones to broadly benefit text embedding learning and
            create positive effects in various scenarios.''
},
``outline'': {
    ``Background'': [
        ``Text embeddings are foundational for semantic textual
            similarity tasks.'',
```

```
                ``Text embeddings benefit downstream tasks like information
                    retrieval and clustering.'',
                ``Text embeddings are particularly useful in LLMs-based
                    applications.''
            ],
            ``Problem and Limitations of Existing Methods'': [
                ``Optimizing cosine functions in STS training leads to
                    gradient vanishing.'',
                ``Cosine saturation zones prevent parameter updates during
                    backpropagation.'',
                ``Models struggle to discern implicit differences in
                    embeddings in saturation zones.'',
                ``STS data often have embedding pairs in saturation zones
                    affecting label clarity.''
            ],
            ``Brief Method Overview and Summary of Main Results'': [
                ``The AoE model decomposes embeddings for handling cosine
                    saturation zones.'',
                ``AoE optimizes angle differences for better similarity
                    learning.'',
                ``AoE introduces a long-text STS dataset from GitHub Issues
                    for evaluation.'',
                ``AoE outperforms baselines on short- and long-text STS
                    tasks.'',
                ``AoE achieves SOTA results on the Massive Text Embedding
                    Benchmark.'',
                ``An ablation study confirms positive contributions from all
                    AoE modules.'',
                ``AoE improves embedding effectiveness in saturation zones.''
            ],
            ``Our Contributions'': [
                ``Investigating cosine saturation zones and optimizing angle
                    differences.'',
                ``Introducing a long-text STS dataset for comprehensive
                    empirical studies.'',
                ``Extensive experiments showing AoE benefits for text
                    embedding learning.''
            ]
        }
}
```

## C  PROMPT FOR GENERATING INTRODUCTION

Table 5 presents two prompt templates in our experiments for guiding LLMs to generate the introduction section, including pure prompt and ELABORATE prompt.

Table 5: **Prompt levels for generating the introduction section**. It includes Pure Prompt and ELAB-ORATE Prompt. Among them, ELABORATE Prompt explicitly requires that large language models be written structurally.

| PROMPT LEVEL | DESCRIPTION |
| --- | --- |
| Pure Prompt | You are an expert in academic paper writing. Please proceed with the academic writing in accordance with the relevant requirements. Please just write the Introduction section of the paper, and do not include any references to other works or authors. The paper should be written in a formal tone and should be suitable for submission to an academic conference. Please write the Introduction section of the academic paper based on the following requirements: 1. Your task is to write the Introduction section of an academic paper based on the given abstract, figures, experimental results tables and baseline abstracts. 2. The language should conform to academic standards, using accurate professional terminology, with a word count controlled between 600 and 1100 words. 3. Please do not write subtitles such as \*\*Background\*\* to show the sturcture of the introduction, the subtitle is not suitable for introduction. Given title: {title} Given abstract: {abstract} Given figures: {figures} Given tables: {tables} Given references (These baseline references only exist in experiments): {baseline_references} Please write a paper Introduction based on the above information. The Introduction should be well-structured, coherent, and follow the conventions of academic writing. Ensure that the introduction is original and does not contain any references to other works or authors. The paper should be written in a formal tone and should be suitable for submission to an academic conference. |
| | *Continued on next page* |

| PROMPT LEVEL | DESCRIPTION |
|---|---|
| ELABORATE Prompt | You are an AI assistant tasked with generating a detailed and well-structured "Introduction" section of a research paper based on the provided title, abstract, and research materials. The abstract of the paper outlines its main objectives, methods, and potential contributions. Effectively integrate the given information to establish a clear research context, articulate the significance of existing gaps, and explicitly highlight the paper's methods and results as well as how it addresses these gaps through its novel contributions, and finally state the contributions. 
 **Important Format Requirements**: 
 - Your response MUST consist of EXACTLY FOUR PARAGRAPHS for the "Introduction". 
 - DO NOT deviate from this four-paragraph structure. 
 - Each paragraph must be between 100 and 150 words, totaling approximately 600 words. 
 **Structure**: 
 1. Paragraph 1: Broad overview of the research area, contextual insights from related materials, significance of the topic. 
 2. Paragraph 2: Specific problem or gap identified, supported by related materials. 
 3. Paragraph 3: Novel contributions of the target paper, including its methods and results, and how it addresses the gaps. 
 4. Paragraph 4: Summary of significance, potential impact, and research purpose. 
 **Style and Content Requirements**: 
 - Maintain a formal academic tone. 
 - Be as coherent and concise as possible, and directly related to the title and abstract. 
 - Use transitional phrases effectively. 
 **Citation Instructions**: 
 - Do not mention any citations. For example, "(Smith et al.)". 
 Target Paper: 
 Title: {title} 
 Abstract: {abstract} 
 Figures: {figures} 
 Tables: {tables} 
 References(These baseline references only exist in experiments): {baseline_references} 
 **Introduction**: |

# D EXAMPLES

The following is the introduction generated by the AutoSurvey and STIG model under the same paper setting. STIG is trained employing the Qwen2.5-7B-Instruct as the backbone model. We provide the original title and abstract along with the generated Introduction. Results of AutoSurvey are shown in Table 6 and Table 7. Results of STIG model are shown in Table 8 and Table 9.

## D.1 AUTOSURVEY: EXAMPLES I

**Title**:

Towards Multi-dimensional Evaluation of LLM Summarization across Domains and Languages

**Abstract**:

Evaluation frameworks for text summarization have evolved in terms of both domain coverage and metrics. However, existing benchmarks still lack domain-specific assessment criteria, remain predominantly English-centric, and face challenges with human annotation due to the complexity of reasoning. To address these, we introduce MSumBench, which provides a multi-dimensional, multi-domain evaluation of summarization in English and Chinese. It also incorporates specialized assessment criteria for each domain and leverages a multi-agent debate system to enhance annotation quality. By evaluating eight modern summarization models, we discover distinct performance patterns across domains and languages. We further examine large language models as summary evaluators, analyzing the correlation between their evaluation and summarization capabilities, and uncovering systematic bias in their assessment of self-generated summaries. Our benchmark dataset is publicly available at https://github.com/DISL-Lab/MSumBench.

Table 6: Generated Introduction by AutoSurvey: Towards Multi-dimensional Evaluation of LLM Summarization across Domains and Languages.

---

**Generated Introduction**

Understanding the necessity of text summarization evaluation involves recognizing the crucial role of summarization techniques in natural language processing (NLP). Over the years, summarization methods have evolved significantly, yet challenges persist, particularly concerning evaluation. Traditional benchmarks often lack domain-specificity and remain predominantly centered on English, posing limitations in capturing diverse contexts and nuances. Moreover, human annotation, a cornerstone of these benchmarks, faces complex reasoning and inter-annotator agreement (IAA) issues. The introduction of MSumBench addresses these gaps by providing a multi-dimensional, multi-domain evaluation for summarization in both English and Chinese. MSumBench employs a multi-agent debate framework to enhance annotation quality and includes domain-specific key-fact categories, offering a more comprehensive and rigorous assessment. These improvements are critical for developing more effective summarization models, especially as large language models increasingly assume roles in both summarization and evaluation tasks.

Unveiling the Limitations of Existing Evaluation Benchmarks highlights several critical issues. Current benchmarks often lack domain-specific assessment criteria, limiting their applicability across diverse domains like news and medical literature, and rely heavily on English-centric evaluations such as mFACE, SummEdits, MFHHD, and UniSumEval, which fail to capture nuances across languages. Additionally, the complexity of human annotation leads to inconsistent inter-annotator agreement, particularly in faithfulness and completeness assessments (IAA ranging from 0.55-0.64 for News and 0.82 for Fact Alignment). Addressing these gaps, MSumBench introduces a multi-dimensional, multi-domain framework that supports domain-specific key-fact evaluations and leverages a multi-agent debate system to enhance annotation quality. This novel approach provides a robust foundation for comprehensive summarization model evaluation, addressing prevailing challenges and offering a more rigorous and nuanced assessment.

---

---

*6*

**Generated Introduction**

Building upon the introduction of MSumBench, this study systematically evaluates summarization systems across English and Chinese domains, employing specialized criteria for each to enhance realism and specificity. Unlike existing benchmarks, MSumBench incorporates a multi-agent debate system to improve annotation quality and reliability. Through the evaluation of eight advanced summarization models, our work reveals distinct performance patterns across various domains and languages, highlighting MSumBench's effectiveness in providing nuanced insights. Notably, our analysis of large language models as evaluators uncovers systematic biases in their self-assessment, underscoring the importance of robust methodologies in summarization research. By addressing the limitations of current evaluation benchmarks, MSumBench offers significant enhancements in annotation quality and evaluation design, providing valuable insights into summarization system capabilities and setting a robust foundation for future research.

Building upon the introduction of MSumBench as a comprehensive evaluation framework that addresses the limitations of existing methods by offering multi-domain support, specialized evaluation criteria, and enhanced annotation quality through a multi-agent debate system, this study systematically examines the performance of eight modern summarization models. By uncovering distinct performance patterns and biases in self-evaluation among Large Language Models (LLMs), MSumBench highlights the complexities inherent in text summarization. The use of LLMs as evaluators reveals systematic discrepancies in their assessment of self-generated summaries, underscoring the need for further refinement in annotation methods and new dimensions of evaluation. This benchmark provides valuable insights for advancing the state of the art in text summarization evaluation, encouraging researchers and practitioners to utilize MSumBench to develop more robust and transparent summarization systems.

Building upon the systematic evaluation of summarization models provided by MSumBench, our framework reveals distinct performance patterns and highlights biases in self-evaluation among Large Language Models (LLMs). By employing LLMs as evaluators, we uncover systematic discrepancies in how they assess their own summaries. This finding underscores the need for refined annotation methods and new dimensions of evaluation. Moving forward, researchers and practitioners are encouraged to utilize MSumBench to advance the state of the art in text summarization evaluation, fostering more robust and transparent summarization systems.

---

## D.2 AUTOSURVEY: EXAMPLES II

**Title**:

Learning to Reason from Feedback at Test-Time

**Abstract**:

Solving complex tasks in a single attempt is challenging for large language models (LLMs). Iterative interaction with the environment and feedback is often required to achieve success, making effective feedback utilization a critical topic. Existing approaches either struggle with length generalization or rely on naive retries without leveraging prior information. In this paper, we introduce FTTT, a novel paradigm that formulates feedback utilization as an optimization problem at test time. Additionally, we propose a learnable test-time optimizer, OpTune, to effectively exploit feedback. Experiments on two LLMs across four reasoning datasets demonstrate that FTTT and OpTune achieve superior scalability and performance.

Table 7: Generated Introduction by AutoSurvey model: Learning to Reason from Feedback at Test-Time.

| **Generated Introduction** |
| --- |
| Managing intricate tasks that necessitate iterative dialogue and feedback poses significant challenges for large language models (LLMs). As these models increasingly engage with complex environments, their effectiveness hinges on the ability to efficiently incorporate feedback into successive interactions. Traditional methods such as sequential revision and parallel sampling struggle with length generalization and lack the ability to self-reflect, leading to suboptimal performance. Moreover, naive retry mechanisms fail to leverage prior knowledge, further hindering progress. To address these limitations, we introduce FTTT (Feedback-Enabled Test-Time Training), a paradigm that formulates feedback utilization as an optimization problem at test time. We also propose OpTune, a learnable optimizer that enhances the exploitation of feedback. Our experimental evaluations on two LLMs across four reasoning datasets underscore the superiority of FTTT and OpTune, demonstrating enhanced scalability and performance compared to existing methods such as revision, self-consistency, and best-of-N approaches. Figures 1 through 4 provide illustrative comparisons and detailed model performance metrics, highlighting the substantial improvements achieved by our proposed framework. |
| Addressing complex tasks that require iterative refinement and continuous feedback presents significant challenges for current methods. Existing schemes such as Revision (Snell et al., 2024), Self-Refine (Madaan et al., 2023), Best-of-N (Brown et al., 2024), and Beam Search (Ow and Morton, 1988) exhibit notable limitations. Revision struggles with length generalization, whereas Self-Refine and Best-of-N fail to efficiently utilize prior iterations, leading to suboptimal performance. Beam Search, despite its efficiency, also falters in length generalization. In contrast, our proposed approach, FTTT, and its key component OpTune, overcome these limitations through a robust optimization framework at test time. Figures 1 and Table 1 illustrate the shortcomings of these baseline methods, particularly in terms of length generalization and the inefficiencies of naive retry strategies. Specifically, FTTT demonstrates superior length generalization and leverages self-reflected feedback for enhanced scalability and performance. Figures 3 and 4 provide further empirical evidence of FTTT's advantages over traditional methods. Next, we introduce Fine-Tuning Through Testing-Time Optimization (FTTT), a novel framework that formulates feedback utilization as an optimization problem at test time. Unlike existing methods, FTTT leverages self-reflected feedback to enhance performance and scalability, thanks to its learnable test-time optimizer, OpTune, which adaptively tunes feedback at each inference step. Our experiments on two large language models across four reasoning datasets demonstrate FTTT's superior performance and scalability, surpassing baseline methods such as revision, self-refine, and best-of-N strategies. Notably, OpTune significantly improves efficiency and accuracy, achieving state-of-the-art results with similar parameter efficiency. Figures 2, 3, and 4 visualize and quantify the effectiveness and scalability of FTTT. |
| Building on the introduction of FTTT and its key component OpTune, we detail how FTTT formulates feedback utilization as an optimization problem at test time, leveraging self-reflected feedback for enhanced performance and scalability. Unlike existing methods such as revision, self-refine, and best-of-N, FTTT introduces a learnable test-time optimizer, OpTune, which dynamically adapts to feedback through adaptive tuning at each inference step. Our experiments on two large language models across four reasoning datasets showcase FTTT's superior performance and scalability, achieving state-of-the-art results with comparable parameter efficiency. Figures 2, 3, and 4 provide visual and quantitative evidence supporting FTTT's effectiveness, while Table 1 highlights its advantages over baseline methods. These findings illustrate how FTTT and OpTune significantly advance the state-of-the-art in feedback utilization for complex reasoning tasks, ensuring more precise adaptation and efficient resource management. This progress is further corroborated by extensive experimental results, detailed in Tables 2 and 3, underscoring the practical applicability of our approach in real-world LLM applications. |

---

*7*

**Generated Introduction**

---

Building upon the contributions of Formulating Test-Time Training (FTTT) and Learnable Test-Time Optimizer (OpTune), these mechanisms significantly advance feedback utilization for large language models (LLMs) tackling complex reasoning tasks. FTTT refines feedback utilization by formulating it as an optimization problem at test time, allowing for more precise adaptation. OpTune introduces a learnable optimizer that efficiently exploits feedback, enhancing both scalability and performance. These improvements are evidenced through comprehensive comparisons with baselines, as demonstrated in Figures 1–4 and Tables 1, 2, and 3. The results show that our approach surpasses traditional methods like sequential revision and beam search in terms of accuracy and resource utilization. These advancements underscore the practical applicability of FTTT and OpTune in real-world LLM applications, setting the stage for more efficient and robust reasoning systems.

---

### D.3 STIG: Examples I

**Title**:

Towards Multi-dimensional Evaluation of LLM Summarization across Domains and Languages

**Abstract**:

Evaluation frameworks for text summarization have evolved in terms of both domain coverage and metrics. However, existing benchmarks still lack domain-specific assessment criteria, remain predominantly English-centric, and face challenges with human annotation due to the complexity of reasoning. To address these, we introduce MSumBench, which provides a multi-dimensional, multi-domain evaluation of summarization in English and Chinese. It also incorporates specialized assessment criteria for each domain and leverages a multi-agent debate system to enhance annotation quality. By evaluating eight modern summarization models, we discover distinct performance patterns across domains and languages. We further examine large language models as summary evaluators, analyzing the correlation between their evaluation and summarization capabilities, and uncovering systematic bias in their assessment of self-generated summaries. Our benchmark dataset is publicly available at https://github.com/DISL-Lab/MSumBench.

**Generated Content**:

⟨STAGE0⟩ Outline for Background:

–Text summarization is a text generation task that aims to condense source text into coherent and abstract summaries.

–The field is divided into abstractive and extractive summarization, with abstractive summarization providing more flexibility and innovation but being challenging due to the need for capturing semantics without literal text. ⟨END0⟩

⟨STAGE1⟩ Contents for Background:

Text summarization is a challenging text generation task that condenses the source text into a few coherent and abstract sentences. It is broadly categorized into abstractive and extractive summarization. Abstractive summarization offers more flexibility and creativity in paraphrasing, condensing, and innovating with the source text, making it a more desirable approach. However, it is significantly challenging as it requires capturing the semantics of the source text without direct literal text. ⟨END1⟩

⟨STAGE2⟩ Outline for Problem and Limitations of Existing Methods:

–Reference summaries in datasets like CNN/Daily Mail and XSum are primarily extractive, making it unclear if models generate extractive summaries due to model limitations or dataset biases.

–Existing datasets lack sufficient domain coverage, and often fail to assess critical dimensions such as faithfulness and completeness.

–Datasets are often language-specific (*e.g.*, English) or task-specific (*e.g.*, news summarization), limiting generalization to other domains or languages.

–Human annotation of summary quality is costly, time-consuming, and faces reliability challenges due to difficulties in reasoning and agreement among annotators.

–Automated evaluation metrics like ROUGE and BERTScore are unreliable because they do not accurately reflect a model's ability to capture semantics and are prone to cheating behaviors like keyword aggregation. ⟨END2⟩

⟨STAGE3⟩ Contents for Problem and Limitations of Existing Methods: Existing datasets typically contain reference summaries that are primarily extractive, making it difficult to determine if models' extractive summaries are due to model limitations or dataset biases. Moreover, existing benchmarks have several drawbacks, including insufficient domain coverage and failure to assess important dimensions such as faithfulness and completeness. Additionally, most datasets are language-specific (*e.g.*, English) or task-specific (*e.g.*, news summarization), limiting the generalization of findings to other domains or languages. Human annotation of summary quality is expensive and time-consuming, and reliability is low due to the complexity of reasoning about multiple quality dimensions. Automatically evaluated metrics, such as ROUGE and BERTScore, are unreliable proxies for summary quality, as they do not ensure the semantic capture of the source text and are susceptible to cheating behaviors like keyword aggregation. ⟨END3⟩

⟨STAGE4⟩ Outline for Brief Method Overview and Summary of Main Results:

–MSumBench is introduced as a multi-dimensional benchmark for evaluating abstractive text summarization models.

–MSumBench covers two domains, News and COVID-19 Research, and is multilingual, supporting summaries in English and Chinese.

–Summaries are evaluated based on dimensions like faithfulness, completeness, and conciseness, with tailored metrics for each domain.

–Human annotators use a multi-agent debate system to reach consensus on summary quality assessments.

–Eight modern summarization models are benchmarked across all dimensions in both languages.

–Performance patterns reveal differences among models and languages in summarization tasks.

–Experiments explore large language models as evaluators of summaries, revealing bias in self-generated summary evaluations. ⟨END4⟩

⟨STAGE5⟩ Contents for Brief Method Overview and Summary of Main Results: To address these issues, we introduce MSumBench, a multi-dimensional evaluation benchmark for abstractive text summarization models. MSumBench covers two diverse domains, namely News and COVID-19 Research, and supports summaries in two languages, English and Chinese. It evaluates summaries based on multiple dimensions, including faithfulness, completeness, and conciseness, with domain-adaptive metrics. An innovative human annotation scheme is employed, where human annotators facilitate a multi-agent debate to arrive at consensus ratings for summary quality. Across eight modern summarization models, we conduct a comprehensive evaluation, assessing their performance across all dimensions in both languages. The results provide insights into distinct performance patterns among models and languages in the context of summarization tasks. Additionally, we explore the feasibility of employing large language models as summary evaluators, providing insights into their evaluation of self-generated summaries and the potential bias involved. ⟨END5⟩

⟨STAGE6⟩ Outline for Our Contributions:

–MSumBench evaluates summarization models across multiple dimensions using adaptive metrics and specialized human annotation processes.

–The benchmark reveals differences in performance among models and languages and sheds light on LLMs' evaluation of self-summarized content. ⟨END6⟩

⟨STAGE7⟩ Contents for Our Contributions:

We introduce a multi-dimensional evaluation benchmark, MSumBench, which assesses summarization models across multiple dimensions using domain-adaptive metrics and specialized human annotation procedures. We evaluate eight modern models in English and Chinese across all benchmark dimensions. Our results shed light on the performance differences among models and languages in summarization tasks and provide insights into large language models' evaluation of self-generated content. ⟨END7⟩

**Generated Introduction**:

Table 8: Generated Introduction by STIG model: Towards Multi-dimensional Evaluation of LLM Summarization across Domains and Languages.

---

**Generated Introduction**

Text summarization is a challenging text generation task that condenses the source text into a few coherent and abstract sentences. It is broadly categorized into abstractive and extractive summarization. Abstractive summarization offers more flexibility and creativity in paraphrasing, condensing, and innovating with the source text, making it a more desirable approach. However, it is significantly challenging as it requires capturing the semantics of the source text without direct literal text.

Existing datasets typically contain reference summaries that are primarily extractive, making it difficult to determine if models' extractive summaries are due to model limitations or dataset biases. Moreover, existing benchmarks have several drawbacks, including insufficient domain coverage and failure to assess important dimensions such as faithfulness and completeness. Additionally, most datasets are language-specific (*e.g.*, English) or task-specific (*e.g.*, news summarization), limiting the generalization of findings to other domains or languages. Human annotation of summary quality is expensive and time-consuming, and reliability is low due to the complexity of reasoning about multiple quality dimensions. Automatically evaluated metrics, such as ROUGE and BERTScore, are unreliable proxies for summary quality, as they do not ensure the semantic capture of the source text and are susceptible to cheating behaviors like keyword aggregation.

To address these issues, we introduce MSumBench, a multi-dimensional evaluation benchmark for abstractive text summarization models. MSumBench covers two diverse domains, namely News and COVID-19 Research, and supports summaries in two languages, English and Chinese. It evaluates summaries based on multiple dimensions, including faithfulness, completeness, and conciseness, with domain-adaptive metrics. An innovative human annotation scheme is employed, where human annotators facilitate a multi-agent debate to arrive at consensus ratings for summary quality. Across eight modern summarization models, we conduct a comprehensive evaluation, assessing their performance across all dimensions in both languages. The results provide insights into distinct performance patterns among models and languages in the context of summarization tasks. Additionally, we explore the feasibility of employing large language models as summary evaluators, providing insights into their evaluation of self-generated summaries and the potential bias involved.

We introduce a multi-dimensional evaluation benchmark, MSumBench, which assesses summarization models across multiple dimensions using domain-adaptive metrics and specialized human annotation procedures. We evaluate eight modern models in English and Chinese across all benchmark dimensions. Our results shed light on the performance differences among models and languages in summarization tasks and provide insights into large language models' evaluation of self-generated content.

---

### D.4 STIG: EXAMPLES II

**Title**:

Learning to Reason from Feedback at Test-Time

**Abstract**:

Solving complex tasks in a single attempt is challenging for large language models (LLMs). Iterative inter-action with the environment and feedback is often required to achieve success, making effective feedback utilization a critical topic. Existing approaches either struggle with length generalization or rely on naive retries without leveraging prior information. In this paper, we introduce FTTT, a novel paradigm that formulates feedback utilization as an optimization problem at test time. Additionally, we propose a learnable test-time optimizer, OpTune, to effectively exploit feedback. Experiments on two LLMs across four reasoning datasets demonstrate that FTTT and OpTune achieve superior scalability and performance.

**Generated Content**:

⟨STAGE0⟩ Outline for Background:

–Large Language Models (LLMs) face challenges in solving complex tasks in a single attempt due to limitations in knowledge retention and reasoning capabilities.

–LLMs can improve their performance by interacting with environments and receiving feedback. ⟨END0⟩

⟨STAGE1⟩ Contents for Background:

Large Language Models (LLMs) often lack the capability to solve complex tasks in a single attempt due to limited knowledge retention and reasoning capabilities. However, LLMs can improve their performance by interacting with the environment and receiving feedback. ⟨END1⟩

⟨STAGE2⟩ Outline for Problem and Limitations of Existing Methods: –Effective utilization of feedback for iterative improvement is a key challenge.

–Existing methods like revision, self-consistency, and beam search do not utilize feedback effectively.

–These methods do not align well with the distribution shift in later attempts, leading to a performance gap.

–Iterative refinement using techniques such as self-instruction and prompt tuning shows promise but lacks optimal feedback utilization.

–Learnable methods for feedback incorporation, such as adapter-based models and LoRA, are limited in scope and do not maximize performance.

–Feedback-based test-time training offers a flexible and powerful approach but faces unresolved challenges. ⟨END2⟩

⟨STAGE3⟩ Contents for Problem and Limitations of Existing Methods: Utilizing feedback effectively to achieve iterative improvement poses a significant challenge. For example, techniques such as revision, self-consistency, and beam search primarily focus on generating multiple candidates and selecting the best one based on predefined metrics. However, they do not utilize feedback effectively for iterative improvement. This approach creates a performance gap when applied in environments with noise or sparse feedback, as it does not align well with the distribution shift in later attempts. Recently, iterative refinement methods, such as self-instruction and prompt tuning, have shown promise in improving task performance. These methods involve using the model's own predictions as feedback and continuously tuning the model parameters or prompts. However, there is still a lack of optimal feedback utilization in these approaches. To address the limitations of parameter-insensitive methods, learnable methods for feedback incorporation have been proposed, such as adapter-based models, LoRA, and full fine-tuning. These methods treat the feedback as

additional data for learning new things rather than for test-time improvements. Adapter-based learning and LoRA are lightweight solutions for incorporating feedback, but they do not maximize performance. Full fine-tuning allows for tuning all parameters of the LLM, but it is not scalable for large models and requires costly retraining each time the model is deployed. Recently, feedback-based test-time training has been proposed, which formulates feedback utilization as a prediction problem at test time. It offers a flexible and powerful approach that can potentially handle any type of feedback and is easily integrated with existing LLMs. However, challenges remain in optimizing the revision generation and effectively utilizing feedback. ⟨END3⟩

⟨STAGE4⟩ Outline for Brief Method Overview and Summary of Main Results: –The proposed FTTT framework uses Test-Time Training for feedback utilization at test time.

–FTTT formulates feedback revision as an optimization problem and leverages a learnable test-time optimizer, OPTUNE, to address feedback utilization.

–OPTUNE is trained jointly with the LLM and serves as a reusable component, reducing computational costs.

–FTTT and OPTUNE together provide a scalable solution for feedback-based learning without requiring retraining the LLM.

–Experiments indicate that FTTT and OPTUNE achieve strong performance results. ⟨END4⟩

⟨STAGE5⟩ Contents for Brief Method Overview and Summary of Main Results: In this paper, we propose a novel framework called FTTT, which leverages Test-Time Training (TTT) for feedback utilization at test time. FTTT formulates feedback revision as an optimization problem and introduces a learnable test-time optimizer, OPTUNE, to solve this problem. By learning the feedback revision, FTTT enables LLMs to better leverage feedback for iterative improvement. The learnable optimizer, OPTUNE, is jointly trained with the LLM during the finetuning phase. Afterward, it can be reused for multiple feedback-based learning tasks, eliminating the need for additional training and dramatically reducing computational costs. Thus, FTTT and OPTUNE empower LLMs with the ability to scale to any testing environment without requiring retraining the LLM. Experiments show that FTTT and OPTUNE achieve impressive performance. ⟨END5⟩

⟨STAGE6⟩ Outline for Our Contributions:

–Propose FTTT, a Test-Time Training-based framework for effective feedback utilization in LLMs.

–Introduce OPTUNE, a learnable optimizer that optimizes feedback revision for improved reasoning.

–Conduct extensive experiments on reasoning tasks, demonstrating the framework's efficacy and scalability. ⟨END6⟩

⟨STAGE7⟩ Contents for Our Contributions:

In summary, our contributions are as follows: • We propose a novel Test-Time Training (TTT)-based framework, called FTTT, for effective feedback utilization in LLMs. • We introduce a learnable optimizer, OPTUNE, which is trained jointly with the LLM to optimize feedback revision at test time. • Extensive experiments on four reasoning tasks demonstrate that our framework enables LLMs to effectively utilize feedback for iterative improvement in a scalable manner. ⟨END7⟩

**Generated Introduction**:

Table 9: Generated Introduction by STIG model: Learning to Reason from Feedback at Test-Time.

**Generated Introduction**

Large Language Models (LLMs) often lack the capability to solve complex tasks in a single attempt due to limited knowledge retention and reasoning capabilities. However, LLMs can improve their performance by interacting with the environment and receiving feedback.

Utilizing feedback effectively to achieve iterative improvement poses a significant challenge. For example, techniques such as revision, self-consistency, and beam search primarily focus on generating multiple candidates and selecting the best one based on predefined metrics. However, they do not utilize feedback effectively for iterative improvement. This approach creates a performance gap when applied in environments with noise or sparse feedback, as it does not align well with the distribution shift in later attempts. Recently, iterative refinement methods, such as self-instruction and prompt tuning, have shown promise in improving task performance. These methods involve using the model's own predictions as feedback and continuously tuning the model parameters or prompts. However, there is still a lack of optimal feedback utilization in these approaches. To address the limitations of parameter-insensitive methods, learnable methods for feedback incorporation have been proposed, such as adapter-based models, LoRA, and full fine-tuning. These methods treat the feedback as additional data for learning new things rather than for test-time improvements. Adapter-based learning and LoRA are lightweight solutions for incorporating feedback, but they do not maximize performance. Full fine-tuning allows for tuning all parameters of the LLM, but it is not scalable for large models and requires costly retraining each time the model is deployed. Recently, feedback-based test-time training has been proposed, which formulates feedback utilization as a prediction problem at test time. It offers a flexible and powerful approach that can potentially handle any type of feedback and is easily integrated with existing LLMs. However, challenges remain in optimizing the revision generation and effectively utilizing feedback.

In this paper, we propose a novel framework called FTTT, which leverages Test-Time Training (TTT) for feedback utilization at test time. FTTT formulates feedback revision as an optimization problem and introduces a learnable test-time optimizer, OPTUNE, to solve this problem. By learning the feedback revision, FTTT enables LLMs to better leverage feedback for iterative improvement. The learnable optimizer, OPTUNE, is jointly trained with the LLM during the finetuning phase. Afterward, it can be reused for multiple feedback-based learning tasks, eliminating the need for additional training and dramatically reducing computational costs. Thus, FTTT and OPTUNE empower LLMs with the ability to scale to any testing environment without requiring retraining the LLM. Experiments show that FTTT and OPTUNE achieve impressive performance.

In summary, our contributions are as follows: • We propose a novel Test-Time Training (TTT)-based framework, called FTTT, for effective feedback utilization in LLMs. • We introduce a learnable optimizer, OPTUNE, which is trained jointly with the LLM to optimize feedback revision at test time. • Extensive experiments on four reasoning tasks demonstrate that our framework enables LLMs to effectively utilize feedback for iterative improvement in a scalable manner.

Table 10: **Generalization performance on CVPR dataset.**

| Method | SS | SR | CC | NQ | QC |
|---|---|---|---|---|---|
| GPT | 0.972 | 0.813 | 0.419 | 28.566 | 1 |
| GPT (ELABORATE) | 0.970 | 0.909 | 0.518 | 32.608 | 1 |
| Pure Prompt | 0.974 | 0.745 | 0.388 | 27.318 | 1 |
| ELABORATE Prompt | 0.971 | 0.741 | 0.351 | 29.722 | 1 |
| Outline Writing | 0.971 | 0.702 | 0.348 | 29.923 | 1 |
| AutoSurvey | 0.967 | 0.616 | 0.295 | 17.796 | 1 |
| FT w/o Stage Writing | 0.980 | 0.789 | 0.442 | 26.682 | 1 |
| Stage Writing w/o FT | 0.969 | 0.681 | 0.398 | 20.565 | 1 |
| Stage Writing FT | 0.978 | 0.808 | 0.441 | 38.308 | 1 |
| STIG (Ours) | 0.976 | 0.831 | 0.459 | 26.978 | 1 |

## D.5 GENERALIZATION PERFORMANCE ON CVPR DATASET

## D.6 EXAMPLES OF GPT-4O GENERATED INTRODUCTIONS

Below are two common mistakes frequently observed in GPT-4o-generated Introductions, such as excessive table enumeration and overstating achievements. We have bolded the portions that do not meet academic standards.

**Generated Introduction**:

Table 11: Generated Introduction by GPT-4o model: Evaluating Design Decisions for Dual Encoder-based Entity Disambiguation.

---

**Generated Introduction (Pure Prompt)**

Entity disambiguation (ED) is a fundamental task in natural language processing, facilitating the linkage of ambiguous mentions within text to their corresponding entities in a knowledge base. A persistent challenge in ED is achieving accurate and efficient matching between text mentions and their candidate labels, a task complicated by the vast number of potential entities and the diverse contexts in which mentions appear. Dual Encoder-based models have emerged as an effective solution to this challenge, embedding both mentions and labels into a shared vector space where a similarity metric is used to predict the correct label association.

This paper explores critical design decisions in the creation of Dual Encoder-based models for entity disambiguation, focusing specifically on their impact on model performance and efficiency. The core decisions under examination include the selection of loss functions, the choice of similarity metrics, the format of label verbalizations, and strategies for negative sampling. These components collectively determine the model's ability to effectively discriminate between correct and incorrect entity associations. Understanding their interactions and optimizing them is essential for enhancing entity disambiguation systems.

We introduce VerbalizED, a document-level Dual Encoder-based entity disambiguation model that functions as a showcase of best practices derived from our evaluation of critical model design decisions. VerbalizED integrates a sophisticated label verbalization format and implements a robust negative sampling strategy. Label verbalization, the process by which entities are represented as embeddings, significantly influences model accuracy. Table 1 exemplifies various components for creating label verbalizations, demonstrating their role in achieving nuanced embedding representations. Subsequent evaluations outlined in Table 2 underscore how enriched verbalization formats, combining titles, descriptions, and categories, markedly improve the F1 score, reinforcing the benefit of comprehensive label representations.

---

Table 11: Generated Introduction by GPT-4o model: Evaluating Design Decisions for Dual Encoder-based Entity Disambiguation. (Continued)

---

**Generated Introduction (Pure Prompt)**

---

Another critical component in the construction of the VerbalizED model is the loss function, which dictates both the training dynamics and the discriminative capacity of the model. Empirical results presented in Table 3 highlight the efficacy of varying span pooling methods under different loss functions—specifically triplet and cross-entropy losses—and illustrate the substantial impact on model performance as observed in F1 scores. The complementary role played by similarity metrics is showcased in Table 4, where a comparative analysis reveals that the choice between cosine, dot product, and Euclidean metrics directly influences the model's accuracy in discerning close embedding pairs.

Negative sampling, a pivotal aspect of training, involves the strategic selection of incorrect label embeddings to enhance model discrimination during training. Table 5 delineates the comparison between various negative sampling methods within the triplet and cross-entropy loss contexts. The evidence suggests that hard negative sampling, particularly with dynamic settings, yields the most favorable improvements in performance metrics, supporting the approach taken in our proposed model.

We further explore innovative variants of VerbalizED, particularly those employing iterative prediction strategies. Figure 2 describes an iterative prediction variant where mentions with initially high similarity are reinserted into the text, thereby refining predictions through enriched contextual embeddings. Table 8 demonstrates the practical impact of this approach with examples showing enhanced prediction accuracy after multiple steps.

Moreover, the frequency of label embedding updates has been identified as an area with considerable impact on the model's efficacy, as seen **in Table 6**. Frequent updates, especially employing an on-the-fly mechanism, dramatically improve performance, suggesting that dynamic adaptation of embeddings during model training is advantageous.

Consistent with the benchmarks, comprehensive experiments on carefully curated datasets, particularly the AIDA-Yago and ZELDA benchmarks, validate our model's superior performance. **As detailed in Table 7**, VerbalizED achieves a new state-of-the-art result on the ZELDA benchmark, with iterative prediction and training variants further pushing the boundaries of achievable accuracy, attesting to the effectiveness of the design decisions incorporated in our model.

The elucidated design choices form the crux of our approach, offering empirical insights that enhance understanding and implementation of Dual Encoder-based entity disambiguation systems. **This work not only advances the field of ED but also provides a robust framework for developing future models with optimized design parameters**. Our exploration of new techniques **paves the way for** ongoing refinement and adaptation in entity disambiguation methodologies, promising improved capabilities in processing knowledge-intensive tasks fundamental to computational linguistics and artificial intelligence.

---

Table 12: Generated Introduction by GPT-4o model: Cropable Knowledge Graph Embedding.

**Generated Introduction (Elaborate Prompt)**

Knowledge Graph Embedding (KGE) is an influential approach within artificial intelligence, focusing on embedding entities and relations from knowledge graphs into continuous vector spaces. This technique enhances numerous AI tasks by facilitating efficient representation and reasoning over complex datasets. The applications of KGE span various domains such as natural language processing, recommender systems, and semantic web technologies, underpinning the backbone of modern AI systems. The accelerating demand for scalable solutions in handling complex relations and vast amounts of data inherent in real-world knowledge graphs highlights the significance of innovations in the KGE domain. Current advancements promise improved efficiency, robustness, and adaptability, making KGE crucial for future AI developments.

Despite its pervasive utilization, KGE encounters several challenges related to dimensional adaptability. Traditional methods necessitate retraining models when different dimensional requirements arise, hampering the flexibility required for diverse scenarios. Such limitations result in increased computational costs and inefficiencies, posing a significant barrier to KGE's widespread application. Consequently, there is a critical need for frameworks that allow flexibility in dimensional embeddings without compromising performance. Existing research materials affirm the complexity of managing and scaling embedding dimensions across varying applications, illustrating a substantial gap in existing methodologies for seamless integration across multiple contexts. Addressing these issues is essential for **leveraging KGE's full potential and enabling its broader applicability in AI-driven tasks.**

This paper introduces a novel framework, MED, designed to efficiently handle varying dimensional needs without retraining from scratch. By permitting direct cropping of sub-models for different dimensions, MED overcomes the dimensional adaptability challenges prevalent in current KGE methods. Innovative mechanisms within MED include mutual learning and evolutionary improvement to bolster low-dimensional models, and dynamic loss weights for a balanced approach to training diverse models. Empirical evaluations demonstrate its effectiveness across four standard datasets and real-world scenarios, along with extensions to the BERT language model. MED significantly improves efficiency and flexibility, offering a transformative method for embedding knowledge graphs in AI applications.

The contributions of this research are both impactful and transformative, addressing pivotal challenges associated with flexible dimensional embeddings in KGE. By enabling efficient and adaptable embedding models, this framework reduces computational demands and facilitates rapid deployment across diverse applications. The potential impact spans improvements in resource utilization, **acceleration of AI-driven processes, and enhancement of modeling complexity within various AI systems**. Ultimately, this work not only fills existing methodological gaps but also establishes a foundation for future explorations into scalable and adaptable KGE frameworks, thereby **fueling advancements in AI technologies.**

Table 13: Ranking of Methods

| id | 1 | 2 | 3 | 4 | 5 | 6 | 7 |
|----|---|---|---|---|---|---|---|
| 47 | STIG | FT w/o Stage Writing | GPT (ELABORATE) | ELABORATE Prompt | GPT | Stage Writing w/o FT | AutoSurvey |
| 50 | STIG | GPT (ELABORATE) | GPT | FT w/o Stage Writing | ELABORATE Prompt | Stage Writing w/o FT | AutoSurvey |
| 139 | FT w/o Stage Writing | GPT | Stage Writing w/o FT | ELABORATE Prompt | AutoSurvey | GPT (ELABORATE) | STIG |
| 147 | STIG | GPT | GPT (ELABORATE) | Stage Writing w/o FT | AutoSurvey | ELABORATE Prompt | FT w/o Stage Writing |
| 1570 | FT w/o Stage Writing | STIG | GPT | GPT (ELABORATE) | AutoSurvey | ELABORATE Prompt | Stage Writing w/o FT |
| 1541 | GPT (ELABORATE) | GPT | ELABORATE Prompt | FT w/o Stage Writing | STIG | Stage Writing w/o FT | AutoSurvey |
| 1539 | STIG | GPT (ELABORATE) | GPT | FT w/o Stage Writing | AutoSurvey | Stage Writing w/o FT | ELABORATE Prompt |
| 1532 | STIG | FT w/o Stage Writing | GPT (ELABORATE) | GPT | AutoSurvey | ELABORATE Prompt | Stage Writing w/o FT |
| 1466 | STIG | FT w/o Stage Writing | GPT (ELABORATE) | GPT | Stage Writing w/o FT | AutoSurvey | ELABORATE Prompt |
| 1429 | FT w/o Stage Writing | Stage Writing w/o FT | GPT (ELABORATE) | STIG | AutoSurvey | GPT | ELABORATE Prompt |
| 1350 | FT w/o Stage Writing | GPT | GPT (ELABORATE) | STIG | ELABORATE Prompt | Stage Writing w/o FT | AutoSurvey |
| 1308 | STIG | FT w/o Stage Writing | Stage Writing w/o FT | GPT (ELABORATE) | ELABORATE Prompt | GPT | AutoSurvey |
| 1302 | STIG | GPT | FT w/o Stage Writing | GPT (ELABORATE) | ELABORATE Prompt | AutoSurvey | Stage Writing w/o FT |
| 1072 | STIG | FT w/o Stage Writing | Stage Writing w/o FT | GPT | GPT (ELABORATE) | AutoSurvey | ELABORATE Prompt |
| 1062 | STIG | ELABORATE Prompt | GPT | FT w/o Stage Writing | GPT (ELABORATE) | Stage Writing w/o FT | AutoSurvey |
| 1041 | STIG | FT w/o Stage Writing | GPT (ELABORATE) | Stage Writing w/o FT | GPT | ELABORATE Prompt | AutoSurvey |
| 1008 | GPT (ELABORATE) | STIG | ELABORATE Prompt | GPT | Stage Writing w/o FT | FT w/o Stage Writing | AutoSurvey |
| 940 | STIG | FT w/o Stage Writing | GPT | GPT (ELABORATE) | AutoSurvey | ELABORATE Prompt | Stage Writing w/o FT |
| 923 | STIG | FT w/o Stage Writing | AutoSurvey | GPT (ELABORATE) | Stage Writing w/o FT | ELABORATE Prompt | GPT |
| 879 | STIG | Stage Writing w/o FT | GPT (ELABORATE) | AutoSurvey | GPT | FT w/o Stage Writing | ELABORATE Prompt |
| 875 | GPT | STIG | GPT (ELABORATE) | ELABORATE Prompt | FT w/o Stage Writing | Stage Writing w/o FT | AutoSurvey |
| 1572 | FT w/o Stage Writing | GPT | GPT (ELABORATE) | ELABORATE Prompt | STIG | AutoSurvey | Stage Writing w/o FT |
| 872 | FT w/o Stage Writing | STIG | GPT (ELABORATE) | ELABORATE Prompt | AutoSurvey | Stage Writing w/o FT | GPT |
| 845 | STIG | FT w/o Stage Writing | GPT | GPT (ELABORATE) | ELABORATE Prompt | Stage Writing w/o FT | AutoSurvey |
| 803 | GPT (ELABORATE) | GPT | AutoSurvey | ELABORATE Prompt | FT w/o Stage Writing | Stage Writing w/o FT | STIG |
| 782 | FT w/o Stage Writing | STIG | ELABORATE Prompt | Stage Writing w/o FT | GPT (ELABORATE) | GPT | AutoSurvey |
| 773 | FT w/o Stage Writing | STIG | ELABORATE Prompt | GPT (ELABORATE) | Stage Writing w/o FT | AutoSurvey | GPT |
| 764 | STIG | FT w/o Stage Writing | ELABORATE Prompt | Stage Writing w/o FT | AutoSurvey | GPT (ELABORATE) | GPT |
| 724 | STIG | GPT | ELABORATE Prompt | GPT (ELABORATE) | Stage Writing w/o FT | AutoSurvey | FT w/o Stage Writing |
| 708 | GPT (ELABORATE) | GPT | ELABORATE Prompt | STIG | AutoSurvey | FT w/o Stage Writing | Stage Writing w/o FT |
| 702 | STIG | FT w/o Stage Writing | GPT (ELABORATE) | GPT | ELABORATE Prompt | Stage Writing w/o FT | AutoSurvey |
| 680 | FT w/o Stage Writing | STIG | GPT | Stage Writing w/o FT | GPT (ELABORATE) | AutoSurvey | ELABORATE Prompt |
| 661 | FT w/o Stage Writing | GPT | Stage Writing w/o FT | GPT (ELABORATE) | STIG | ELABORATE Prompt | AutoSurvey |
| 658 | STIG | GPT (ELABORATE) | FT w/o Stage Writing | Stage Writing w/o FT | ELABORATE Prompt | GPT | AutoSurvey |
| 634 | STIG | FT w/o Stage Writing | GPT | ELABORATE Prompt | GPT (ELABORATE) | AutoSurvey | Stage Writing w/o FT |
| 615 | FT w/o Stage Writing | GPT | GPT (ELABORATE) | STIG | ELABORATE Prompt | Stage Writing w/o FT | AutoSurvey |
| 591 | FT w/o Stage Writing | STIG | GPT | ELABORATE Prompt | GPT (ELABORATE) | AutoSurvey | Stage Writing w/o FT |
| 579 | STIG | GPT | FT w/o Stage Writing | ELABORATE Prompt | Stage Writing w/o FT | GPT (ELABORATE) | AutoSurvey |
| 540 | STIG | FT w/o Stage Writing | ELABORATE Prompt | GPT (ELABORATE) | GPT | Stage Writing w/o FT | AutoSurvey |
| 409 | FT w/o Stage Writing | STIG | ELABORATE Prompt | GPT | GPT (ELABORATE) | AutoSurvey | Stage Writing w/o FT |
| 338 | GPT | FT w/o Stage Writing | STIG | ELABORATE Prompt | GPT (ELABORATE) | AutoSurvey | Stage Writing w/o FT |
| 324 | FT w/o Stage Writing | ELABORATE Prompt | GPT | GPT (ELABORATE) | Stage Writing w/o FT | AutoSurvey | STIG |
| 289 | STIG | GPT | AutoSurvey | FT w/o Stage Writing | Stage Writing w/o FT | GPT (ELABORATE) | ELABORATE Prompt |
| 270 | STIG | FT w/o Stage Writing | GPT | Stage Writing w/o FT | AutoSurvey | ELABORATE Prompt | GPT (ELABORATE) |
| 262 | STIG | FT w/o Stage Writing | GPT | GPT (ELABORATE) | ELABORATE Prompt | AutoSurvey | Stage Writing w/o FT |
| 249 | FT w/o Stage Writing | STIG | GPT | GPT (ELABORATE) | Stage Writing w/o FT | ELABORATE Prompt | AutoSurvey |
| 223 | FT w/o Stage Writing | STIG | GPT | Stage Writing w/o FT | GPT (ELABORATE) | ELABORATE Prompt | AutoSurvey |
| 180 | FT w/o Stage Writing | STIG | GPT | ELABORATE Prompt | GPT (ELABORATE) | AutoSurvey | Stage Writing w/o FT |
| 169 | STIG | FT w/o Stage Writing | Stage Writing w/o FT | AutoSurvey | GPT (ELABORATE) | GPT | ELABORATE Prompt |
| 158 | FT w/o Stage Writing | STIG | GPT (ELABORATE) | GPT | ELABORATE Prompt | Stage Writing w/o FT | AutoSurvey |

