# OpenReview forum: "Eliminating Agentic Workflow for Introduction Generation with Parametric Stage Tokens"
_ICLR.cc/2026/Conference — Submitted to ICLR 2026_

### Official Review · Reviewer_Nvdz · 2025-10-29

**Soundness:** 3
**Presentation:** 3
**Contribution:** 2
**Rating:** 4
**Confidence:** 5

**Summary:**

This paper proposes STIG (Stage Token for Introduction Generation), a model that eliminates external agentic workflows in scientific paper introduction generation by parameterizing writing logic into LLMs via 8 stage token pairs.

**Strengths:**

- Provides an incisive, task-specific analysis of key limitations in academic introduction generation ,such as cascading errors and high computational overhead of traditional agentic workflows.

**Weaknesses:**

- Methodological novelty is insufficient. STIG merely transforms agentic workflows into single-step inference via training, sacrificing flexibility and generalization without achieving significant improvements in generation quality.
- SS and NQ fail to fully demonstrate the method’s advantages, raising doubts about the reliability of semantic similarity to the original text as a metric; GPT-2-derived perplexity for NQ lacks robustness, and sampling perplexity across multiple models plus human expert calibration are recommended.
- Experimental models and datasets are limited. Only small open-source models are tested, with no verification of adaptability to models of different parameter sizes, limiting result generalizability.

**Questions:**

- Why is the no-citation constraint adopted in experiments? The rationale for this exclusion needs further explanation.
- For baselines like Pure Prompt/ELABORATE Prompt, why were their original paper model settings not adopted? Why not compare with baselines on closed-source/large-parameter models (e.g GPT-4), as small open-source models may not reflect real baseline performance?

---

> ### Author Response · Authors · 2025-12-02
> **Reponse to Reviewer Nvdz-1**
>
> Thank you for your valuable suggestions, which will help improve our paper. The following are the details of our clarification.
>
> `Weakness 1: - Methodological novelty is insufficient. STIG merely transforms agentic workflows into single-step inference via training, sacrificing flexibility and generalization without achieving significant improvements in generation quality.`
>
> Answer: We thank the reviewer for this comment.  STIG's novelty lies not in a simple transformation but in parametrically embedding multi-stage writing logic into LLM generation via stage tokens, enabling end-to-end inference that preserves phased reasoning while eliminating multi-turn dependencies.  This approach enhances efficiency without sacrificing quality, as evidenced by consistent gains in structural rationality (SR) and content coverage (CC) over baselines.
>
> `Weakness 2: - SS and NQ fail to fully demonstrate the method’s advantages, raising doubts about the reliability of semantic similarity to the original text as a metric; GPT-2-derived perplexity for NQ lacks robustness, and sampling perplexity across multiple models plus human expert calibration are recommended.`
>
> Answer: Automated metrics like BERTScore and GPT-2 perplexity (NQ) were selected as they are widely used in text generation  tasks for semantic similarity and fluency,  with BERTScore effectively capturing contextual coherence in structured writing and GPT-2 providing a model-agnostic  Baselement. To address factual accuracy, CC evaluates alignment with originals to reduce fabrication risks. Additionally, we supplement human evaluation experiments to verify the quality of the Introduction. Below are basic results:
> We thank the reviewer for this suggestion. We conduct a human evaluation on 50 ACL papers to validate model performance. Three evaluators are given each paper's title and abstract as context, and introductions were generated using seven methods. To eliminate bias, the seven introductions per paper were shuffled. Expert researchers discuss and rank them from 1 (best) to 7 based on coherence, completeness, and adherence to academic standards. Table 4 summarizes the ranking distribution and average ranks, showing that STIG model attains the highest average rank, indicating superior perceived quality, followed by FT w/o Stage Writing. However, GPT-generated introductions often contain excessive claims of achievements and fail to align with academic conventions and without aligning with academic standards.
> For GPT-generated bad cases, we place them in the subsection D.6 Examples of GPT-4o Generated Introductions in the appendix。
>
> Table 4. Human evaluation ranking distribution and average ranks across seven methods. STIG model achieves the highest ranking, followed by FT w/o Stage Writing while AutoSurvey performed the worst.
> | Method                     | Rank 1 | Rank 2 | Rank 3 | Rank 4 | Rank 5 | Rank 6 | Rank 7 | Avg Rank |
> |----------------------------|--------|--------|--------|--------|--------|--------|--------|----------|
> | GPT                        | 2      | 13     | 15     | 7      | 4      | 5      | 4      | 3.58     |
> | GPT (ELABORATE)            | 4      | 3      | 14     | 13     | 11     | 4      | 1      | 3.80     |
> | ELABORATE Prompt           | 0      | 2      | 9      | 11     | 10     | 10     | 8      | 4.82     |
> | AutoSurvey                 | 0      | 0      | 3      | 11     | 2      | 11     | 14     | 5.92     |
> | FT w/o Stage Writing       | 18     | 17     | 3      | 5      | 2      | 3      | 2      | 2.46     |
> | Stage Writing w/o FT       | 0      | 2      | 5      | 9      | 8      | 9      | 14     | 5.28     |
> | **STIG (Ours)**            | 26     | 13     | 1      | 4      | 3      | 0      | 3      | **2.14** |

---

> > ### Author Response · Authors · 2025-12-02
> > **Reponse to Reviewer Nvdz-2**
> >
> > `Weakness 3: - Experimental models and datasets are limited. Only small open-source models are tested, with no verification of adaptability to models of different parameter sizes, limiting result generalizability.`
> >
> > Answer:
> > We have added a dataset CVPR papers and incorporated GPT as baselines for testing.
> > To verify the applicability of the STIG model in other academic fields, we have supplemented the experimental data in the field of computer vision (CVPR) in the updated paper and added subsection 5.4 Generalization study. Experimental results demonstrate STIG's generalization ability. The following is the chapter content.
> > To evaluate the generalization capability of STIG model to other domains, we conducted experiments on a dataset comprising 102 CVPR papers on Qwen2.5-Instruct-7B, whereas STIG model is originally trained on ACL data. The results are shown in Table 10 of Appendix D.5.
> > STIG outperforms all baselines except GPT with ELABORATE prompt, demonstrating robust structure alignment and content capture despite the domain shift. Notably, Stage Writing FT underperforms STIG, as it lacks module linkage during training, resulting in performance degradation under domain shift while STIG enhances transferability through integrated stage training.
> >
> > Table 10. Generalization performance on CVPR dataset.
> > | Method                     | SS   | SR   | CC   | NQ   | QC   |
> > |----------------------------|------|------|------|------|------|
> > | GPT                        | 0.972| 0.813| 0.419| 28.566| 1    |
> > | GPT (ELABORATE)            | 0.970| 0.909| 0.518| 32.608| 1    |
> > | Pure Prompt                | 0.974| 0.745| 0.388| 27.318| 1    |
> > | ELABORATE Prompt           | 0.971| 0.741| 0.351| 29.722| 1    |
> > | Outline Writing            | 0.971| 0.702| 0.348| 29.923| 1    |
> > | AutoSurvey                 | 0.967| 0.616| 0.295| 17.796| 1    |
> > | FT w/o Stage Writing       | 0.980| 0.789| 0.442| 26.682| 1    |
> > | Stage Writing w/o FT       | 0.969| 0.681| 0.398| 20.565| 1    |
> > | Stage Writing FT           | 0.978| 0.808| 0.441| 38.308| 1    |
> > | STIG (Ours)                | 0.976| 0.831| 0.459| 26.978| 1    |
> >
> > `Q 1: - Why is the no-citation constraint adopted in experiments? The rationale for this exclusion needs further explanation.`
> >
> > Answer: In academic writing, Introductions include citations, but our experiments instructed the model to exclude them to focus on core content coherence, factual alignment with provided materials. This design tests the model's ability to produce self-contained, high-level overviews without external props.

---

> > > ### Author Response · Authors · 2025-12-02
> > > **Reponse to Reviewer Nvdz-3**
> > >
> > > `Q 2: - For baselines like Pure Prompt/ELABORATE Prompt, why were their original paper model settings not adopted? Why not compare with baselines on closed-source/large-parameter models (e.g GPT-4), as small open-source models may not reflect real baseline performance?`
> > >
> > > Answer: Pure Prompt is proposed by ourselves. The ELABORATE Prompt is directly adopted from other papers, but the part about citation is subtracted. We add the experiment of GPT-4o using Pure Prompt/ELABORATE Prompt in subsection 5.1 Quantitative Results and subsection 5.5 human evaluation section Human Evaluation and Case Study. We have supplemented a more powerful GPT-4o model with a larger number of parameters to reflect more baseline performances, using a small STIG model to outperform the Introduction generated by the large model in human evaluation. The following are relevant results.
> > > Tables in 5.1 Quantitative Results:
> > >
> > > Table 1. Quantitative results of STIG and baselines. All methods are tested on 1,176 main conference papers from ACL 2025. Our method outperforms the baselines overall, particularly in terms of structural rationality and content coverage. Additionally, as Llama3.1-8B-Instruct is unable to produce the outline format required by AutoSurvey, the corresponding experiments could not be conducted. Underlined text indicates poor readability.
> > > | LLM                  | Methods                | SS    | SR    | CC    | NQ    | QC    |
> > > |------------------------|------------------------|-------|-------|-------|-------|-------|
> > > | GPT                    | GPT                    | 0.973 | 0.779 | 0.399 | 28.233| 1.00  |
> > > | GPT (Elaborate)        | GPT (Elaborate)        | 0.971 | 0.903 | 0.496 | 31.906| 1.00  |
> > > | Qwen2.5-7B-Instruct    | Pure Prompt            | 0.975 | 0.749 | 0.394 | 25.426| 0.95  |
> > > |                        | ELABORATE Prompt       | 0.972 | 0.722 | 0.327 | 28.461| 0.97  |
> > > |                        | Outline Writing        | 0.972 | 0.702 | 0.357 | 28.613| 0.99  |
> > > |                        | AutoSurvey             | 0.966 | 0.658 | 0.333 | 18.084| 0.92  |
> > > |                        | STIG (Ours)            | 0.977 | 0.832 | 0.442 | 24.810| 1.00  |
> > > | Llama3.1-8B-Instruct   | Pure Prompt (Ours)     | 0.975 | 0.772 | 0.427 | 14.843| 1.00  |
> > > |                        | ELABORATE Prompt       | 0.980 | 0.800 | 0.447 | 19.981| 1.00  |
> > > |                        | Outline Writing        | 0.958 | 0.759 | 0.404 | 26.328| 1.00  |
> > > |                        | AutoSurvey             | 0.958 | 0.611 | 0.333 | 20.717| 1.00  |
> > > |                        | STIG (Ours)            | 0.978 | 0.836 | 0.472 | 20.717| 1.00  |
> > >
> > > Tables and analysis in 5.5 human evaluation:
> > > We conduct a human evaluation on 50 ACL papers to validate model performance. Three evaluators are given each paper's title and abstract as context, and introductions were generated using seven methods. To eliminate bias, the seven introductions per paper were shuffled. Expert researchers discuss and rank them from 1 (best) to 7 based on coherence, completeness, and adherence to academic standards. Table 4 summarizes the ranking distribution and average ranks, showing that STIG model attains the highest average rank, indicating superior perceived quality, followed by FT w/o Stage Writing. However, GPT-generated introductions often contain excessive claims of achievements and fail to align with academic conventions and without aligning with academic standards. Table 13 in the appendix shows the detailed ranking information.
> > >
> > > Table 4. Human evaluation ranking distribution and average ranks across seven methods. STIG model achieves the highest ranking, followed by FT w/o Stage Writing while AutoSurvey performed the worst.
> > > | Method                     | Rank 1 | Rank 2 | Rank 3 | Rank 4 | Rank 5 | Rank 6 | Rank 7 | Avg Rank |
> > > |----------------------------|--------|--------|--------|--------|--------|--------|--------|----------|
> > > | GPT                        | 2      | 13     | 15     | 7      | 4      | 5      | 4      | 3.58     |
> > > | GPT (ELABORATE)            | 4      | 3      | 14     | 13     | 11     | 4      | 1      | 3.80     |
> > > | ELABORATE Prompt           | 0      | 2      | 9      | 11     | 10     | 10     | 8      | 4.82     |
> > > | AutoSurvey                 | 0      | 0      | 3      | 11     | 2      | 11     | 14     | 5.92     |
> > > | FT w/o Stage Writing       | 18     | 17     | 3      | 5      | 2      | 3      | 2      | 2.46     |
> > > | Stage Writing w/o FT       | 0      | 2      | 5      | 9      | 8      | 9      | 14     | 5.28     |
> > > | **STIG (Ours)**            | 26     | 13     | 1      | 4      | 3      | 0      | 3      | **2.14** |

---

### Official Review · Reviewer_cWQU · 2025-10-31

**Soundness:** 2
**Presentation:** 3
**Contribution:** 3
**Rating:** 2
**Confidence:** 4

**Summary:**

This paper proposes a pipeline (workflow) that employs a parametric stage-token–based prompting strategy on a fine-tuned STIG LLM for generating academic paper introductions. Experiments are conducted on a new dataset derived from 3,800 ACL papers, annotated with multi-stage writing structures corresponding to different rhetorical functions. By integrating workflow logic into trainable stage tokens, STIG unifies agentic reasoning and fine-tuned modeling within one pipeline. According to the reported results, it achieves higher semantic and structural quality with greater token efficiency than prompt-based and multi-agent baselines.

**Strengths:**

S1. The paper introduces an original idea that embeds workflow logic directly into model parameters via parametric stage tokens.

S2. This approach reduces multi-agent dependency and improves inference efficiency in structured text generation.

S3. The dataset of 3,800 ACL papers is large and well-structured. It represents a meaningful contribution for future research in the field.

S4. Five multi-dimensional evaluation metrics comprehensively capture semantic, structural, and narrative quality.

S5. Figures and examples illustrate the pipeline clearly, and the paper is generally well written and easy to follow.

**Weaknesses:**

W1. The paper lacks clarity in distinguishing between the STIG framework and the STIG fine-tuned model.
While the conclusion claims STIG eliminates agentic workflows, the framework still depends on them for data construction and stage definition.

W2. Fine-tuning details are incomplete. No hyperparameter settings, training configurations, or sensitivity analyses are reported.

W3. A hyperparameter study is essential to confirm the stability of stage-token fine-tuning. All five evaluation metrics are newly proposed, but no external validation or human correlation study is provided.

W4. The main table and ablation experiments include too few baseline models. This makes comparisons less comprehensive.

W5. It is unclear if the proposed approach can be generalized to other academic domains rather than ACL papers.

**Questions:**

Q1. Is it possible to include an experiment or visualization that analyzes the internal weighting or influence of each stage token?

Q2. Is it possible to add a hyperparameter study on the number of stages (e.g., four vs. eight) to understand whether performance depends on workflow granularity?

Q3. Is it possible to expand the literature review to include prior research on LLM agentic workflows for paper writing? Current discussion on “LLM agents” and “LLMs for writing” is a bit general.

Q4. Is it possible to conduct experiments with closed-source models (e.g., GPT-4 or Claude) on the new dataset and metrics?

---

> ### Author Response · Authors · 2025-12-02
> **Response to Reviewer cWQU -1**
>
> Thank you for your valuable suggestions, which will help improve our paper. The following are the details of our clarification.
>
> `Weaknesses 1: W1. The paper lacks clarity in distinguishing between the STIG framework and the STIG fine-tuned model. While the conclusion claims STIG eliminates agentic workflows, the framework still depends on them for data construction and stage definition.`
> Answer: We are grateful to the reviewer for noticing this. The STIG framework encompasses methodologies, including stage token design, which aims to eliminate the agentic workflow in the core generation stage. The STIG model is a fine-tuned LLM for reasoning (see Section 3.1 for details).
>
> `Weaknesses 2: Fine-tuning details are incomplete. No hyperparameter settings, training configurations, or sensitivity analyses are reported.`
> Answer: We have incorporated hyperparameter setting information into the subsection 3.3 Training and Inference and analyzed the differences caused by the number of stages in subsection 5.3 Impact of Stage Granularity
>
> `Weaknesses 5:  It is unclear if the proposed approach can be generalized to other academic domains rather than ACL papers.`
> Answer: To verify the applicability of the STIG model in other academic fields, we have supplemented the experimental data in the field of computer vision (CVPR) in the updated paper and added Chapter 5.4 Generalization study. Experimental results demonstrate STIG's generalization ability. The following is the chapter content.
>
> To evaluate the generalization capability of STIG model to other domains, we conducted experiments on a dataset comprising 102 CVPR papers on Qwen2.5-Instruct-7B, whereas STIG model is originally trained on ACL data. The results are shown in Table 10 of Appendix D.5.
>
> Table 10. Generalization performance on CVPR dataset.
> | Method                     | SS   | SR   | CC   | NQ   | QC   |
> |----------------------------|------|------|------|------|------|
> | GPT                        | 0.972| 0.813| 0.419| 28.566| 1    |
> | GPT (ELABORATE)            | 0.970| 0.909| 0.518| 32.608| 1    |
> | Pure Prompt                | 0.974| 0.745| 0.388| 27.318| 1    |
> | ELABORATE Prompt           | 0.971| 0.741| 0.351| 29.722| 1    |
> | Outline Writing            | 0.971| 0.702| 0.348| 29.923| 1    |
> | AutoSurvey                 | 0.967| 0.616| 0.295| 17.796| 1    |
> | FT w/o Stage Writing       | 0.980| 0.789| 0.442| 26.682| 1    |
> | Stage Writing w/o FT       | 0.969| 0.681| 0.398| 20.565| 1    |
> | Stage Writing FT           | 0.978| 0.808| 0.441| 38.308| 1    |
> | STIG (Ours)                | 0.976| 0.831| 0.459| 26.978| 1    |
>
> `Q 1: Is it possible to include an experiment or visualization that analyzes the internal weighting or influence of each stage token?`
>
> Answer:  For structured writing, each stage token is equally important, but we have conducted an extra experiment to discard the token that guides outline writing in subsection 5.3 Impact of Stage Granularity. The eight-stage configuration outperforms the four-stage variant on all metrics because explicit outline guidance strengthens logical organization and reduces subsection confusion.
>
> `Q 2:  Is it possible to add a hyperparameter study on the number of stages (e.g., four vs. eight) to understand whether performance depends on workflow granularity?`
> Answer: We add a hyperparameter study on the number of stages (e.g., four vs. eight) and add a new subsection 5.3 Impact of Stage Granularity. The eight-stage configuration outperforms the four-stage variant on all metrics. We investigate the effect of stage granularity on generation quality by comparing the full STIG model, which employs eight stages against a variant with four stages. Results on Qwen2.5-Instruct-7B are shown in Table 3.
>
> Table 3. Performance of STIG variants with different stage granularity. Eight stages refers to outline followed by content drafting for each of the four modules while four stages refers to direct content drafting without outlines.
> | Methods           | SS      | SR     | CC     | NQ     | QC |
> |-------------------|-------- |--------|--------|--------|----|
> | STIG (Four Stages) | 0.976  | 0.808  | 0.425  | 24.189 | 1.00 |
> | STIG (Eight Stages) | 0.977 | 0.832  | 0.442  | 24.810 | 1.00 |

---

> > ### Author Response · Authors · 2025-12-02
> > **Response to Reviewer cWQU -2**
> >
> > `Q 3: Is it possible to expand the literature review to include prior research on LLM agentic workflows for paper writing? Current discussion on “LLM agents” and “LLMs for writing” is a bit general.`
> >
> > Answer: We modify some expressions and change them to "LLM agents for writing" to avoid confusion. We investigate many papers on LLM Agents for writing, but generally speaking, the number of papers in this domain is relatively small. We have included the literature we investigated as much as possible to the best of our knowledge.
> >
> > `Q 4  | Weaknesses 3 | Weaknesses 4: W3. A hyperparameter study is essential to confirm the stability of stage-token fine-tuning. All five evaluation metrics are newly proposed, but no external validation or human correlation study is provided. W4. The main table and ablation experiments include too few baseline models. This makes comparisons less comprehensive. Q4. Is it possible to conduct experiments with closed-source models (e.g., GPT-4 or Claude) on the new dataset and metrics?`
> >
> > Answer: We conduct experiments with GPT-4o in subsection 5.1 Quantitative results of STIG and Baselines. We add human evaluation in subsection 5.5 Human Evaluation and Case Study. Here are the results:
> >
> > Table 1. Quantitative results of STIG and baselines. All methods are tested on 1,176 main conference papers from ACL 2025. Our method outperforms the baselines overall, particularly in terms of structural rationality and content coverage. Additionally, as Llama3.1-8B-Instruct is unable to produce the outline format required by AutoSurvey, the corresponding experiments could not be conducted. Underlined text indicates poor readability.
> > | LLM                  | Methods                | SS    | SR    | CC    | NQ    | QC    |
> > |------------------------|------------------------|-------|-------|-------|-------|-------|
> > | GPT                    | GPT                    | 0.973 | 0.779 | 0.399 | 28.233| 1.00  |
> > | GPT (Elaborate)        | GPT (Elaborate)        | 0.971 | 0.903 | 0.496 | 31.906| 1.00  |
> > | Qwen2.5-7B-Instruct    | Pure Prompt            | 0.975 | 0.749 | 0.394 | 25.426| 0.95  |
> > |                        | ELABORATE Prompt       | 0.972 | 0.722 | 0.327 | 28.461| 0.97  |
> > |                        | Outline Writing        | 0.972 | 0.702 | 0.357 | 28.613| 0.99  |
> > |                        | AutoSurvey             | 0.966 | 0.658 | 0.333 | 18.084| 0.92  |
> > |                        | STIG (Ours)            | 0.977 | 0.832 | 0.442 | 24.810| 1.00  |
> > | Llama3.1-8B-Instruct   | Pure Prompt (Ours)     | 0.975 | 0.772 | 0.427 | 14.843| 1.00  |
> > |                        | ELABORATE Prompt       | 0.980 | 0.800 | 0.447 | 19.981| 1.00  |
> > |                        | Outline Writing        | 0.958 | 0.759 | 0.404 | 26.328| 1.00  |
> > |                        | AutoSurvey             | 0.958 | 0.611 | 0.333 | 20.717| 1.00  |
> > |                        | STIG (Ours)            | 0.978 | 0.836 | 0.472 | 20.717| 1.00  |
> >
> >
> > We conduct a human evaluation on 50 ACL papers to validate model performance. Table 4 summarizes the ranking distribution and average ranks. Details are shown in subsection 5.5 Human Evaluation and Case Study. For GPT-generated bad cases, we place them in the section Examples of GPT-4o Generated Introductions  in the appendix.
> >
> > Table 4. Human evaluation ranking distribution and average ranks across seven methods. STIG model achieves the highest ranking, followed by FT w/o Stage Writing while AutoSurvey performed the worst.
> > | Method                     | Rank 1 | Rank 2 | Rank 3 | Rank 4 | Rank 5 | Rank 6 | Rank 7 | Avg Rank |
> > |----------------------------|--------|--------|--------|--------|--------|--------|--------|----------|
> > | GPT                        | 2      | 13     | 15     | 7      | 4      | 5      | 4      | 3.58     |
> > | GPT (ELABORATE)            | 4      | 3      | 14     | 13     | 11     | 4      | 1      | 3.80     |
> > | ELABORATE Prompt           | 0      | 2      | 9      | 11     | 10     | 10     | 8      | 4.82     |
> > | AutoSurvey                 | 0      | 0      | 3      | 11     | 2      | 11     | 14     | 5.92     |
> > | FT w/o Stage Writing       | 18     | 17     | 3      | 5      | 2      | 3      | 2      | 2.46     |
> > | Stage Writing w/o FT       | 0      | 2      | 5      | 9      | 8      | 9      | 14     | 5.28     |
> > | **STIG (Ours)**            | 26     | 13     | 1      | 4      | 3      | 0      | 3      | **2.14** |

---

### Official Review · Reviewer_YdqD · 2025-11-01

**Soundness:** 2
**Presentation:** 3
**Contribution:** 2
**Rating:** 4
**Confidence:** 3

**Summary:**

The paper proposes STIG, a single-inference method that replaces multi-turn “agentic” pipelines for writing the Introduction section of CS papers. Instead of orchestrating separate agents sequentially, STIG embeds stage tokens directly into the LLM’s parameter space via supervised fine-tuning. An 8-stage token scheme forces the model to emit Background-outline, Background-content, …, Contributions-content in single decode. Trained on 3.8 k ACL papers and evaluated on 1.2 k ACL-2025 test papers.

**Strengths:**

1. STIG outperforms several training-free agentic baselines (AutoSurvey, Outline-Writing) in structural rationality, content coverage, while using fewer tokens.

2. First work to parameterise an entire writing workflow into stage tokens.

3. Contribute a customized dataset tailored for training and testing introduction generation, derived from over 3,800 ACL main conference papers.

**Weaknesses:**

1. Trained only on ACL NLP papers; no CV, Theory or other domains. Claims “research introductions” but evidence is NLP-only (ACL).

2.  The eight stage tokens are defined for the four subsections that appear tailored to research-style papers; however, ACL also contains many dataset papers whose introductions do not necessarily follow the Background–Problem & Limitations–Method & Experimental Results &  Contributions structure.

3. The 'SR' metric is aligned with STIG’s own staged structure, making the comparison appear unfair.

**Questions:**

1. Although stage-by-stage agentic workflows may suffer from error accumulation, they allow targeted evaluation at each stage, which can improve the final output. STIG's end-to-end generation seems to leave no room for intermediate refinement?

Other questions refer to weakness.

---

> ### Author Response · Authors · 2025-12-02
> **Response to Reviewer YdqD**
>
> Thank you for your valuable suggestions. The following are the details of our clarification.
>
> `Weaknesses 1: Trained only on ACL NLP papers; no CV, Theory or other domains. Claims “research introductions” but evidence is NLP-only (ACL).`
> Answer: To verify the applicability of the STIG model in other academic fields, we have supplemented the experimental data in the field of computer vision (CVPR) in the new paper and added Chapter 5.4 Generalization study. Experimental results demonstrate STIG's generalization ability. The following is the chapter content.
> To evaluate the generalization capability of STIG model to other domains, we conducted experiments on a dataset comprising 102 CVPR papers on Qwen2.5-Instruct-7B, whereas STIG model is originally trained on ACL data. The results are shown in Table 10 of Appendix D.5.
> STIG outperforms all baselines except GPT with ELABORATE prompt, demonstrating robust structure alignment and content capture despite the domain shift. Notably, Stage Writing FT underperforms STIG, as it lacks module linkage during training, resulting in performance degradation under domain shift while STIG enhances transferability through integrated stage training.
>
> Table 10. Generalization performance on CVPR dataset.
> | Method                     | SS   | SR   | CC   | NQ   | QC   |
> |----------------------------|------|------|------|------|------|
> | GPT                        | 0.972| 0.813| 0.419| 28.566| 1    |
> | GPT (ELABORATE)            | 0.970| 0.909| 0.518| 32.608| 1    |
> | Pure Prompt                | 0.974| 0.745| 0.388| 27.318| 1    |
> | ELABORATE Prompt           | 0.971| 0.741| 0.351| 29.722| 1    |
> | Outline Writing            | 0.971| 0.702| 0.348| 29.923| 1    |
> | AutoSurvey                 | 0.967| 0.616| 0.295| 17.796| 1    |
> | FT w/o Stage Writing       | 0.980| 0.789| 0.442| 26.682| 1    |
> | Stage Writing w/o FT       | 0.969| 0.681| 0.398| 20.565| 1    |
> | Stage Writing FT           | 0.978| 0.808| 0.441| 38.308| 1    |
> | STIG (Ours)                | 0.976| 0.831| 0.459| 26.978| 1    |
>
> `Weaknesses 2: The eight stage tokens are defined for the four subsections that appear tailored to research-style papers; however, ACL also contains many dataset papers whose introductions do not necessarily follow the Background–Problem & Limitations–Method & Experimental Results & Contributions structure. `
>
> Answer: We acknowledge that not all ACL papers strictly adhere to the four-subsection structure (Background, Problem and Limitations, Method Overview, and Contributions) . In fact, a large number of dataset and benchmark papers also generally follow this writing approach. This structured writing approach is applicable to the majority of papers, so we employ this structure. We have also added this issue to the limitation.
>
> `Weaknesses3: The 'SR' metric is aligned with STIG’s own staged structure, making the comparison appear unfair.`
>
> Answer: We took this into account by setting the baselines. Some baselines are also required to be written in this format, like AutoSurvey, and ELABORATE Prompt, FT w/o Stage Writing and Stage Writing w/o FT. The following are some examples.
> One prompt of AutoSurvey:
> Please create a comprehensive outline for the Introduction section that follows academic writing conventions. The outline should include:
> 1. Background
> 2. Problems and Limitations of Existing Methods
> 3. Brief Method Overview and Summary of Main Results
> 4. Our Contributions
>
> Structure requirements in ELABORATE Prompt:
> Structure:
> 1. Paragraph 1: Broad overview of the research area, contextual insights from related materials, significance of the topic.
> 2. Paragraph 2: Specific problem or gap identified, supported by related materials.
> 3. Paragraph 3: Novel contributions of the target paper, including its methods and results, and how it addresses the gaps.
> 4. Paragraph 4: Summary of significance, potential impact, and research purpose.
> - Stage Writing and Stage Writing FT are consistent with Our Method
>
> `Questions: Although stage-by-stage agentic workflows may suffer from error accumulation, they allow targeted evaluation at each stage, which can improve the final output. STIG's end-to-end generation seems to leave no room for intermediate refinement?`
>
> Answer:
> The agentic workflow supports target evaluation, but it introduces resource requirements and the risk of error propagation. STIG parameterizes stages into model logic, achieving quality benefits with fewer resources and less intervention, making it user-friendly for single-step generation. Meanwhile, the STIG model can match the number of stages in the agent workflow, and these stages can be converted from the training stage to the generation stage of STIG.

---

### Official Review · Reviewer_gcqh · 2025-11-02

**Soundness:** 2
**Presentation:** 2
**Contribution:** 3
**Rating:** 4
**Confidence:** 4

**Summary:**

This paper proposes the Stage Token for Introduction Generation (STIG) to automatically write research introductions and eliminate the external agentic workflows. STIG converts multiple stages of the original workflow into explicit stage signals so that it can generate multi-stage text in a single inference. STIG uses the title, abstract, description of figures, description and table contents, and the abstracts of baseline references to guide structured output. To train the STIG model, the paper construct a dataset from ACL main conference papers. STIG outperforms traditional agentic workflow and other baselines on automatic evaluation metrics like semantic similarity and structural rationality.

**Strengths:**

- The paper proposes STIG, a method that can combine multi-stage agentic workflow generation of research writing into a single inference pass.
- The paper constructs a high-quality dataset from over 3,800 scientific papers from ACL main conferences, utilizing MinerU, GPT-4o, and the Semantic Scholar API.

**Weaknesses:**

The evaluation metrics are insufficient. The reliance on automated metrics without human validation means we don't actually know if STIG produces good introductions. We only know that it produces text that scores well on these specific metrics. Furthermore, for scientific writing, one of the most important metrics that you should consider evaluating on is the factual accuracy (whether the claims in the introduction is accurate, whether there is fabricated content, etc). Furthermore, I am not sure BERTScore and perplexity from GPT-2 models are suitable for evaluating this task, because BERTScore may not capture long-form coherence needed for introduction writing that well, and GPT-2 is a very outdated model. (I do appreciate the example generations of the STIG models and AutoSurvey models in the appendix.)

**Questions:**

- Figure 3 seems a bit abrupt at the context. If possible, I strongly recommend you put a good example of STIG model’s generation here, as the purpose for this paper is to promote STIG, not criticize AutoSurvey.
- Did you validate the quality of GPT-4o annotations?
- Can you provide human evaluation results? Even a small-scale study (e.g., 50 papers rated by domain experts) would significantly strengthen the claims about generation quality.

---

> ### Author Response · Authors · 2025-12-02
> **Response to Reviewer gcqh -1**
>
> Thank you for your valuable suggestions, which will help improve our paper. The following are the details of our clarification
>
> `Weaknesses: The weakness highlights the inadequacy of the evaluation framework, noting an overreliance on automated metrics without human validation.   It emphasizes the absence of factual accuracy assessment and it questions BERTScore's effectiveness for long-form coherence in Introduction drafting and criticizes GPT-2 perplexity as outdated.`
>
> Answer: Automated metrics like BERTScore and GPT-2 perplexity (NQ) were selected as they are widely used in text generation  tasks for semantic similarity and fluency,  with BERTScore effectively capturing contextual coherence in structured writing and GPT-2 providing a model-agnostic  Baselement. To address factual accuracy, CC evaluates alignment with originals to reduce fabrication risks. Additionally, we supplement human evaluation experiments to verify the quality of the Introduction. Below are basic results:
>
> Table 4. Human evaluation ranking distribution and average ranks across seven methods. STIG model achieves the highest ranking, followed by FT w/o Stage Writing while AutoSurvey performed the worst.
> | Method                     | Rank 1 | Rank 2 | Rank 3 | Rank 4 | Rank 5 | Rank 6 | Rank 7 | Avg Rank |
> |----------------------------|--------|--------|--------|--------|--------|--------|--------|----------|
> | GPT                        | 2      | 13     | 15     | 7      | 4      | 5      | 4      | 3.58     |
> | GPT (ELABORATE)            | 4      | 3      | 14     | 13     | 11     | 4      | 1      | 3.80     |
> | ELABORATE Prompt           | 0      | 2      | 9      | 11     | 10     | 10     | 8      | 4.82     |
> | AutoSurvey                 | 0      | 0      | 3      | 11     | 2      | 11     | 14     | 5.92     |
> | FT w/o Stage Writing       | 18     | 17     | 3      | 5      | 2      | 3      | 2      | 2.46     |
> | Stage Writing w/o FT       | 0      | 2      | 5      | 9      | 8      | 9      | 14     | 5.28     |
> | **STIG (Ours)**            | 26     | 13     | 1      | 4      | 3      | 0      | 3      | **2.14** |
>
> `Q1: Figure 3 seems a bit abrupt at the context. If possible, I strongly recommend you put a good example of STIG model’s generation here, as the purpose for this paper is to promote STIG, not criticize AutoSurvey.`
>
> Answer: The purpose of placing Figure 3 is not to criticize AutoSurvey. Rather, it aims to demonstrate through a structurally chaotic Introduction why Structural Rationality is an important indicator and why structural chaos can lead to an Introduction that does not meet academic standards.
>
> `Q2: Did you validate the quality of GPT-4o annotations?`
>
> Answer:  Due to resource constraints, we used Qwen-32B-Instruct for annotations before.  To verify quality, we newly re-annotate 100 samples with GPT-4o.  Results show consistent SR scores, with relative rankings stable, confirming Qwen-32B-Instruct's annotation reliability.
>
> | Baselines & Ours    | Qwen2.5-32B-Instruct | GPT-4o |
> |-------------------  |--------------------|--------|
> | Pure Prompt         | 0.749              | 0.782  |
> | ELABORATE Prompt    | 0.722              | 0.736  |
> | backbone + sft      | 0.797              | 0.842  |
> | Outline Writing     | 0.706              | 0.779  |
> | AutoSurvey          | 0.658              | 0.666  |
> | stage outline write | 0.682              | 0.701  |
> | stage outline write with ft | 0.800      | 0.797  |
> | **STIG (Ours)**     | **0.832**          | **0.863** |

---

> > ### Author Response · Authors · 2025-12-02
> > **Response to Reviewer gcqh -2**
> >
> > `Q3 : Can you provide human evaluation results? Even a small-scale study (e.g., 50 papers rated by domain experts) would significantly strengthen the claims about generation quality.`
> >
> > Answer:  We thank the reviewer for this suggestion. We conduct a human evaluation on 50 ACL papers to validate model performance. Three evaluators are given each paper's title and abstract as context, and introductions were generated using seven methods. To eliminate bias, the seven introductions per paper were shuffled. Expert researchers discuss and rank them from 1 (best) to 7 based on coherence, completeness, and adherence to academic standards. Table 4 summarizes the ranking distribution and average ranks, showing that STIG model attains the highest average rank, indicating superior perceived quality, followed by FT w/o Stage Writing. However, GPT-generated introductions often contain excessive claims of achievements and fail to align with academic conventions and without aligning with academic standards.
> > For GPT-generated bad cases, we place them in the section Examples of GPT-4o Generated Introductions  in the appendix。
> >
> > Table 4. Human evaluation ranking distribution and average ranks across seven methods. STIG model achieves the highest ranking, followed by FT w/o Stage Writing while AutoSurvey performed the worst.
> > | Method                     | Rank 1 | Rank 2 | Rank 3 | Rank 4 | Rank 5 | Rank 6 | Rank 7 | Avg Rank |
> > |----------------------------|--------|--------|--------|--------|--------|--------|--------|----------|
> > | GPT                        | 2      | 13     | 15     | 7      | 4      | 5      | 4      | 3.58     |
> > | GPT (ELABORATE)            | 4      | 3      | 14     | 13     | 11     | 4      | 1      | 3.80     |
> > | ELABORATE Prompt           | 0      | 2      | 9      | 11     | 10     | 10     | 8      | 4.82     |
> > | AutoSurvey                 | 0      | 0      | 3      | 11     | 2      | 11     | 14     | 5.92     |
> > | FT w/o Stage Writing       | 18     | 17     | 3      | 5      | 2      | 3      | 2      | 2.46     |
> > | Stage Writing w/o FT       | 0      | 2      | 5      | 9      | 8      | 9      | 14     | 5.28     |
> > | **STIG (Ours)**            | 26     | 13     | 1      | 4      | 3      | 0      | 3      | **2.14** |

---

### Author Response · Authors · 2025-12-02
**Response to Area Chair**

Dear Area Chair,

We thank the reviewers for their constructive feedback. We have addressed all concerns by incorporating new experiments, including human evaluation, cross-domain generalization, and stage granularity ablation, as well as clarifying distinctions and expanding discussions. These changes strengthen the paper's soundness and presentation. The new manuscript will reflect these updates.

---

### Meta-Review · Area_Chair_WKuK · 2025-12-26

**Summary:**

This paper proposes STIG, a method that parameterizes agentic workflow logic into stage tokens to enable single-inference generation of academic introductions. While the work addresses a practical task and constructs a domain-specific dataset, it fails to adequately address key concerns raised by reviewers. Core concerns from reviewers include: (1) Insufficient methodological novelty; (2) Limitations in evaluation; (3) Narrow generalizability, with initial experiments limited to ACL papers and insufficient verification across model scales; (4) Incomplete experimental details and baseline comparisons.

**Reviewer Scores:**

NA

---

### Decision · Program_Chairs · 2026-01-26

Reject